# A Natural Polyphenol, Chlorogenic Acid, Attenuates Obesity-Related Metabolic Disorders in Male Rats via miR-146a-IRAK1-TRAF6 and NRF2-Mediated Antioxidant Pathways

**DOI:** 10.3390/biom15081086

**Published:** 2025-07-27

**Authors:** Rashid Fahed Alenezi, Adel Abdelkhalek, Gehad El-Sayed, Ioan Pet, Mirela Ahmadi, El Said El Sherbini, Daniela Pușcașiu, Ahmed Hamed Arisha

**Affiliations:** 1Biochemistry and Molecular Biology Department, Faculty of Veterinary Medicine, Mansoura University, Mansoura 35516, Egypt; 2Department of Food Safety, Hygiene and Technology, Faculty of Veterinary Medicine, Badr University in Cairo, Badr City 11829, Egypt; 3Department of Biotechnology, Faculty of Bioengineering of Animals Resources, University of Life Sciences “King Mihai I” from Timisoara, 300645 Timisoara, Romania; 4Institute of Applied Biotechnology, 300645 Timisoara, Romania; 5Faculty of Medicine, Victor Babes University of Medicine and Pharmacy Timisoara, 2 Eftimie Murgu Street, 300041 Timisoara, Romania; 6ANAPATMOL Research Center, Victor Babes University of Medicine and Pharmacy Timisoara, 2 Eftimie Murgu Street, 300041 Timisoara, Romania; 7Department of Animal Physiology and Biochemistry, Faculty of Veterinary Medicine, Badr University in Cairo, Badr City 11829, Egypt; 8Department of Physiology, Faculty of Veterinary Medicine, Zagazig University, Zagazig 44519, Egypt

**Keywords:** chlorogenic acid, metabolic syndrome, high-fat diet, inflammation, oxidative stress, adipose tissue, microRNA, adipose inflammation, appetite regulation

## Abstract

Chronic high-fat diet (HFD) feeding in male rats causes significant metabolic as well as inflammatory disturbances, including obesity, insulin resistance, dyslipidemia, liver and kidney dysfunction, oxidative stress, and hypothalamic dysregulation. This study assessed the therapeutic effects of chlorogenic acid (CGA), a natural polyphenol, administered at 10 mg and 100 mg/kg/day for the last 4 weeks of a 12-week HFD protocol. Both CGA doses reduced body weight gain, abdominal circumference, and visceral fat accumulation, with the higher dose showing greater efficacy. CGA improved metabolic parameters by lowering fasting glucose and insulin and enhancing lipid profiles. CGA suppressed orexigenic genes (Agrp, NPY) and upregulated anorexigenic genes (POMC, CARTPT), suggesting appetite regulation in the hypothalamus. In abdominal white adipose tissue (WAT), CGA boosted antioxidant defenses (SOD, CAT, GPx, HO-1), reduced lipid peroxidation (MDA), and suppressed pro-inflammatory cytokines including TNF-α, IFN-γ, and IL-1β, while increasing the anti-inflammatory cytokine IL-10. CGA modulated inflammatory signaling via upregulation of miR-146a and inhibition of IRAK1, TRAF6, and NF-κB. It also reduced apoptosis by downregulating p53, Bax, and Caspase-3, and restoring Bcl-2. These findings demonstrate that short-term CGA administration effectively reverses multiple HFD-induced impairments, highlighting its potential as an effective therapeutic for obesity-related metabolic disorders.

## 1. Introduction

Obesity has become a significant public health concern of the 21st century, with extensive consequences for worldwide morbidity and mortality. According to the World Health Organization in 2023, globally, more than 650 million adults are classified as obese, and this number continues to rise at an alarming rate across all age groups and socioeconomic strata [1]. Obesity is not merely a condition of excess body weight, but a complex, multifactorial disorder characterized by chronic low-grade inflammation, insulin resistance, dyslipidemia, and systemic metabolic dysregulation [2,3]. This growing epidemic not only contributes directly to the development of type 2 diabetes mellitus (T2DM), cardiovascular diseases, and non-alcoholic fatty liver disease (NAFLD), but also exacerbates the risk of several cancers and neurodegenerative conditions [3]. The pathophysiology of obesity is closely intertwined with excessive caloric intake, particularly from diets rich in saturated fats and refined sugars, which collectively promote energy imbalance and visceral fat accumulation [4].

Among dietary factors, high-fat diets have been strongly linked to developing obesity as well as its associated complications. Chronic consumption of HFDs disrupts metabolic homeostasis by altering lipid metabolism, promoting adipocyte hypertrophy, and inducing inflammatory signaling within visceral adipose tissue (VAT) [5,6]. High-fat diet (HFD)-induced obesity is closely associated with metabolic syndrome, a cluster of interrelated conditions including NAFLD, cardiovascular diseases, and T2DM, largely driven by inflammation and oxidative stress [7,8]. The above changes result in the secretion of pro-inflammatory cytokines, such as interleukin-6 (IL-6), interleukin-1β (IL-1β), and tumor necrosis factor-alpha (TNF-α), that exacerbate systemic inflammation and disrupt insulin signaling pathways in the adipose tissue, muscle, and liver [9]. Furthermore, HFD-induced obesity has been shown to affect the central nervous system, particularly within the hypothalamus, where it disrupts the balance between orexigenic (appetite-stimulating) and anorexigenic (appetite-suppressing) neuropeptides such as cocaine- and amphetamine-regulated transcript (CARTPT), pro-opiomelanocortin (POMC), agouti-related peptide (Agrp), and neuropeptide Y (NPY) [10]. This disruption contributes to hyperphagia, reduced satiety, and further exacerbation of weight gain.

Besides metabolic and neuroendocrine disruptions, high-fat-diet-induced obesity also exacerbates oxidative stress by excessively producing reactive oxygen species (ROS) while concurrently depleting intrinsic antioxidant defenses. Oxidative stress is integral to the progression of obesity-related conditions [7,11]. The redox-sensitive transcription factor nuclear factor kappa beta (NF-κB) is activated under conditions of oxidative stress, leading to upregulation of inflammatory genes and perpetuation of the inflammatory cycle in VAT and other metabolically active tissues [8,12]. Moreover, oxidative damage to lipids, proteins, and DNA contributes to cellular dysfunction and apoptosis, particularly in metabolically stressed adipocytes, further impairing tissue integrity and function [13].

With rising urbanization, sedentary lifestyles, and increased consumption of calorie-dense foods, obesity has become a major driver of the global burden of metabolic syndrome, necessitating urgent exploration of effective therapeutic interventions without considerable adverse effects [4]. In recent years, natural bioactive compounds derived from plant-based sources have received considerable consideration for their potential impact on several metabolic pathways as well as improving health outcomes in obesity and related disorders [14,15]. Among these, CGA—a hydroxycinnamic acid derivative found abundantly in coffee, apples, pears, and certain vegetables—has demonstrated promising anti-obesity, anti-inflammatory, antioxidant, and antidiabetic properties in both preclinical and clinical studies [16,17]. While these findings highlight the unique benefits of CGA, it is important to note that other polyphenolic compounds—such as resveratrol, quercetin, and epigallocatechin gallate (EGCG)—have also demonstrated comparable anti-obesity, antioxidant, and anti-inflammatory effects through modulation of similar molecular pathways, including NF-κB inhibition and Nrf2 activation.

CGA exerts its beneficial effects through several interconnected mechanisms. CGA has been demonstrated to impede glucose absorption in the gut and augment glucose uptake in skeletal muscle through enhancing AMP-activated protein kinase (AMP) [18,19]. CGA influences gut microbiota, fostering the proliferation of advantageous bacterial strains that generate short-chain fatty acids and decrease intestinal permeability, thereby alleviating systemic inflammation and metabolic endotoxemia [20,21].

In addition to its metabolic benefits, CGA demonstrates significant anti-inflammatory properties; CGA downregulates the expression of pro-inflammatory cytokines, such as TNF-α, IL-1β, and IL-6, via impeding the NF-κB signaling pathway [16,22]. This effect is especially crucial in the topic of obesity, where persistent inflammation in abdominal visceral adipose tissue (VAT) induces insulin resistance and metabolic decline. Notably, miR-146a, a microRNA known to negatively regulate inflammatory responses by targeting IRAK1 and TRAF6, key components of the Toll-like receptor (TLR) signaling cascade [23,24], may help to restore immune homeostasis and reduce adipose tissue inflammation.

Moreover, CGA demonstrates robust antioxidant properties, which are crucial in combating the oxidative stress associated with obesity. CGA scavenges free radicals directly and enhances endogenous antioxidant defenses by activating the Nrf2/Keap1 pathway, which regulates the expression of antioxidant enzymes like glutathione peroxidase (GPx), catalase (CAT), and superoxide dismutase (SOD) [25,26]. The activation of this pathway not only reduces oxidative damage but also protects against tissue injury and apoptosis in metabolically active organs, including the liver and kidneys. Indeed, studies have shown that CGA treatment significantly decreases markers of oxidative stress and lipid peroxidation, malondialdehyde (MDA), and restores total antioxidant capacity (TAC) in diet-induced obesity in animal models [27].

Despite the accumulating evidence supporting the therapeutic potential of CGA, the precise molecular mechanisms underlying its effects on HFD-induced metabolic dysfunction remain incompletely understood. Specifically, the impact of CGA on adipose tissue apoptosis, hypothalamic signaling, and miRNA-mediated regulation of inflammation requires further exploration. Thus, the present study aimed to explore the comprehensive effects of CGA oral administration at various doses on the metabolic parameters, hormonal profiles, lipid homeostasis, hepatic and renal function, inflammation, oxidative stress, and gene expression in an HFD-induced obesity male rat model. We hypothesized that CGA would attenuate weight gain, improve insulin sensitivity, reduce systemic inflammation, and modulate key molecular pathways in the hypothalamus and abdominal inguinal WAT.

## 2. Materials and Methods

### 2.1. Animals

Forty male Sprague Dawley (SD) rats, averaging 200 ± 10 g in body weight, were obtained from the Laboratory Animal Unit at Badr University in Cairo (BUC), and allowed to acclimate for two weeks before starting the experimental protocol. Rats were maintained at a standard light/dark cycle and temperature conditions, provided with either a commercial normal chow meal or an HFD (60 kcal% from lard [28], see Table 1) ad libitum, and had unrestricted access to water. The diet-induced obesity (DIO) rats were subjected to a 50:50 high-fat and normal diet for two weeks before being switched to a 100% HFD until the end of the study [29]. All experimental procedures involving animals were conducted in strict accordance with the Animal Research: Reporting of In Vivo Experiments (ARRIVE) guidelines (Appendix A) to ensure high standards of ethical conduct and scientific integrity. The study was approved by the Institutional Animal Care and Use Committee (IACUC) of Mansoura University, Egypt (Approval No: Ph.D 121/2021 on 15 October 2021).

### 2.2. Experimental Design

Randomly, rats were allocated to four experimental groups (N = 10 per group) as shown in Table 2 and Figure 1: (1) control group maintained on a standard chow diet for 12 weeks; (2) HFD group supplied with a HFD (60 kcal% from lard) [28] for the entire 12-week period to induce obesity and metabolic dysfunction; (3) HFD + low-dose CGA group, which received HFD for 12 weeks but supplemented with CGA at a dose of 10 mg/kg/day via oral gavage during the final 4 weeks; and (4) HFD + high-dose CGA group, which followed the same HFD protocol but received a higher dose of CGA (100 mg/kg/day) during the last four weeks. The doses of CGA (10 mg/kg/day and 100 mg/kg/day) were selected based on previous studies demonstrating efficacy in rodent models and pilot experiments conducted during preliminary studies [30,31]. The delayed administration of CGA was designed to mimic a therapeutic intervention after metabolic disruption had been established through 8 weeks of HFD feeding. Weekly throughout the study, both food intake and body weight were measured.

### 2.3. Tissue and Blood Sampling

At the end of the experiment, rats of all groups were allowed to fast but had free access to water overnight. Animals were euthanized by cervical dislocation followed by exsanguination after the last exercise session. Blood samples were collected in a BD Vacutainer PST II Tube, allowed to clot at room temperature, followed by centrifugation at 3000× *g* for 20 min. Serum samples were collected and preserved at −20 °C until used for the hormonal assay. Abdominal circumference, body weight, and length of rats were measured, and BMI was calculated (BMI = weight (g)/length^2^ (cm^2^)) [32,33]. Blood samples were collected without anticoagulants for serum separation. The separated serum was then kept at −20 °C until needed. The abdominal inguinal WAT and the hypothalamus samples, carefully dissected from each brain tissue of the euthanized rats as previously described [34], were immediately placed in liquid N_2_ and stored at −80 °C until further molecular analysis. The hypothalamus was dissected from each brain tissue, following the technique described by Glowinski and Iversen [35]. It was defined anteriorly to the rostral end of the optic chiasm, superiorly by the top of the third ventricle, laterally by the fornices, and posteriorly by the mammillary bodies.

### 2.4. Biochemical Analysis

A blood glucose meter (URiGHT, New Taipei City, Taiwan) was used to measure blood glucose levels. Commercially available ELISA kits were used to estimate the levels of insulin, leptin, and ghrelin (MyBioSource, USA) using a DNM-9602 ELISA reader (DRAWELL, Chongqing, China), according to the manufacturer’s instructions. HOMA-IR was calculated as HOMA-IR = (fasting insulin (µU/mL) × fasting glucose (mg/dL))/405. The serum concentrations of AST, ALP, ALT, uric acid, BUN, and creatinine were measured using kits from Spectrum Kits (Egyptian Company for Biotechnology, Cairo, Egypt) [36]. Oxidative stress was assessed by measuring malondialdehyde (MDA) levels using the thiobarbituric acid reactive substances (TBARS) assay, which serves as a reliable indicator of lipid peroxidation. Total antioxidant capacity (TAC) was determined using a colorimetric assay kit according to the manufacturer’s instructions (Spectrum Kits, Egypt). [37,38].

### 2.5. Real-Time PCR Analysis

Total RNA was extracted from hypothalamic and abdominal inguinal white adipose tissue samples as previously reported [34]. Total RNA extraction yielded between 2 and 3 μg of RNA per sample, as measured by NanoDrop spectrophotometry. These quantities were sufficient for reverse transcription and real-time PCR analysis without the need for RNA amplification. RNAs were reverse transcribed using either 500 ng of total RNA (for mRNA) with a high-capacity cDNA reverse transcription kit (iNtRON Biotechnology, Republic of Korea) or 10 ng (for miRNA) with the miScript II Reverse Transcription Kit (Qiagen, Santa Clarita, CA, USA) using a Veriti 96-well thermal cycler (Applied Biosystems, USA). The Stem loop RT and miR144-specific primers, in conjunction with the universal reverse primer, were developed using the UPL probe-based stem-loop quantitative PCR assay design software v3.4 (http://genomics.dote.hu:8080/mirnadesigntool), informed by mature microRNA sequences obtained from the miRNA database at https://www.mirbase.org/ (accessed on 1 May 2024) [39]. Gene expression analysis was performed using specific primers for the target genes (Table 3) provided by Sangon Biotech (Beijing, China) in a real-time RT-PCR assay, executed on aMx3005P Real-Time PCR System (Agilent Stratagene, USA) employing Maxima SYBR Green qPCR Master Mix (2×) with ROX Solution (Thermo Fisher Scientific, USA).

Gene expression was measured using the 2^−ΔΔCt^ technique and normalized to the reference housekeeping gene Gapdh. Gene expression alterations were measured as a fold change relative to the control [40]. Quantification of the expression levels of various miRNAs was performed using stem-loop RT-qPCR for miRNA analysis [41].

### 2.6. Statistical Analysis

Statistical analysis was conducted using GraphPad Prism 9 software (GraphPad Software Inc., USA), which was also utilized for graph generation. Prior to statistical analysis, data were tested for normality using the Shapiro–Wilk test and for homogeneity of variances using Levene’s test. Data meeting these assumptions were analyzed using one-way ANOVA followed by post hoc Tukey’s test. Data are presented as means ± SEM. A *p*-value less than 0.05 is considered statistically significant.

## 3. Results

### 3.1. Metabolic and Anthropometric Effects of HFD and CGA Administration

Chronic HFD feeding induced significant increases in body weight, adiposity, and metabolic dysregulation compared to control animals, while administration with CGA mitigated these effects (Figure 2A–F). Final day body weight was markedly increased in the HFD group (*p* < 0.0001 vs. control), indicating robust weight gain associated with HFD consumption. Both doses of CGA (10 mg and 100 mg) significantly reduced body weight gain (*p* < 0.0001 vs. HFD). Regarding body length, HFD-fed rats showed a significant decrease in comparison to controls (*p* < 0.0001), suggesting developmental or metabolic impairment under HFD conditions. Both doses of CGA (10 mg and 100 mg) significantly improved body length (*p* < 0.05 vs. HFD, *p* < 0.0001 vs. HFD, respectively). BMI, calculated as final body weight normalized to body length squared, was significantly elevated in the HFD-fed rats (*p* < 0.0001 vs. control), reflecting disproportionate fat accumulation relative to body size. This increase was significantly reduced by CGA treatment (*p* < 0.001 vs. HFD). Abdominal circumference, a proxy for visceral obesity, was also significantly higher in HFD-fed rats (*p* < 0.001 vs. control), and this was reduced by both doses of CGA (*p* < 0.01 vs. HFD). Similarly, both absolute and relative abdominal WAT weights were markedly elevated in the HFD-fed rats (*p* < 0.0001 and *p* < 0.05 vs. control), demonstrating excessive visceral fat deposition. Treatment with CGA significantly decreased absolute WAT mass at both doses (*p* < 0.001 and *p* < 0.0001 vs. HFD, respectively), whereas only administration of 100 mg CGA significantly reduced relative abdominal WAT weight (*p* < 0.05 vs. HFD).

### 3.2. Effects of HFD and CGA on Metabolic Hormones

Chronic HFD feeding significantly altered metabolic hormone levels compared to control animals, indicating impaired glucose homeostasis and adipokine signaling, while CGA administration mitigated these disturbances (Figure 3A–E). Serum insulin levels were significantly elevated in HFD-fed rats (*p* < 0.01 vs. control), consistent with metabolic dysregulation associated with obesity. Both doses of CGA (10 mg and 100 mg) significantly reduced serum insulin (*p* < 0.05 and *p* < 0.01 vs. HFD, respectively), suggesting improved insulin sensitivity. Similarly, fasting blood glucose (FBG) was significantly higher in the HFD-fed rats (*p* < 0.0001 vs. control), reflecting glucose intolerance. Administration with CGA led to a significantly reduced fasting glucose level (*p* < 0.001 vs. HFD). HOMA-IR was significantly higher in the HFD-fed rats (*p* < 0.0001 vs. control). Administration with CGA led to a significantly reduced HOMA-IR (*p* < 0.0001 vs. HFD). Serum leptin, an adipokine involved in appetite regulation and energy balance, was significantly elevated in HFD-fed rats (*p* < 0.001 vs. control), indicating leptin resistance and excessive adiposity. Both doses of CGA effectively lowered leptin levels (*p* < 0.05 and *p* < 0.01 vs. HFD, respectively), pointing to a regulatory role in adipokine secretion and fat mass control. In contrast, serum ghrelin, a hunger-stimulating hormone, showed a significant elevation under HFD conditions (*p* < 0.05 vs. control). Ghrelin levels were partially restored by CGA treatment at the 100 mg dose (*p* < 0.05 vs. HFD). Finally, serum adiponectin was significantly reduced in HFD-fed rats (*p* < 0.05 vs. control), consistent with impaired metabolic function. This decrease was reversed by CGA at the 100 mg dose (*p* < 0.05 vs. HFD).

### 3.3. Effects of HFD and CGA on Lipid Profile

Chronic HFD feeding significantly disrupted lipid homeostasis, as evidenced by marked elevations in total cholesterol, TAG, LDL-c, and VLDL-c, along with reduced HDL-c level, while such dyslipidemic effects were effectively reduced by CGA administration (Figure 4A–E). Total cholesterol levels were significantly elevated in HFD-fed rats in comparison with control rats (*p* < 0.0001), indicating impaired cholesterol metabolism. Both doses of CGA (10 mg and 100 mg) significantly reduced total cholesterol levels, with the higher dose showing the most pronounced effect (*p* < 0.001 and *p* < 0.0001 vs. HFD, respectively). Similarly, triglyceride (TAG) levels were significantly elevated in the HFD-fed rats (*p* < 0.0001 vs. control), consistent with hepatic lipid accumulation and metabolic dysfunction. CGA treatment significantly lowered TAG concentrations, particularly at the 100 mg dose (*p* < 0.0001 vs. HFD). LDL-c was significantly elevated in HFD-fed animals (*p* < 0.0001 vs. control), reflecting enhanced atherogenic risk. This increase was significantly mitigated by both doses of CGA (*p* < 0.0001 vs. HFD). VLDL-c followed a similar trend, showing a significant elevation in the HFD-fed rats (*p* < 0.0001 vs. control), which was significantly reduced with CGA treatment, especially at the 100 mg dose (*p* < 0.0001 vs. HFD). In contrast, HDL-c, associated with cardiovascular protection, was significantly suppressed in HFD-fed rats (*p* < 0.0001 vs. control). Administration with CGA restored HDL-c levels (*p* < 0.05 and *p* < 0.01 vs. HFD, respectively).

### 3.4. Effects of HFD and CGA on Hepatic and Renal Function Biomarkers

Chronic HFD feeding significantly elevated serum markers of hepatic and renal dysfunction compared to control animals, indicating organ stress, while CGA administration attenuated these effects (Figure 5A–F). Alanine aminotransferase (ALT) levels, a sensitive indicator of liver damage, were markedly increased in HFD-fed rats (*p* < 0.0001 vs. control), suggesting hepatic injury. Both doses of CGA significantly reduced ALT concentrations (*p* < 0.001 vs. HFD, respectively). Similarly, aspartate aminotransferase (AST), another marker of hepatocellular damage, was significantly elevated in HFD-fed rats (*p* < 0.001 vs. control). Oral administration with CGA showed a significant reduction in AST levels (*p* < 0.001 vs. HFD). Alkaline phosphatase (ALP), an enzyme associated with liver and biliary dysfunction, was also significantly raised in HFD-fed rats (*p* < 0.05 vs. control). Administration with CGA significantly lowered ALP levels (*p* < 0.05 vs. HFD), indicating improved hepatic function. Renal function biomarkers were similarly affected by HFD feeding. Creatinine levels were higher in HFD-fed rats (*p* < 0.001 vs. control), suggesting potential kidney impairment. Both doses of CGA reduced creatinine levels (*p* < 0.01 vs. HFD). Blood urea nitrogen (BUN), another marker of renal function, was significantly increased under HFD conditions (*p* < 0.001 vs. control), which was significantly attenuated via CGA administration (*p* < 0.01 vs. HFD). Finally, uric acid was also elevated in HFD-fed rats (*p* < 0.001 vs. control). CGA administration significantly lowered uric acid levels (*p* < 0.01 vs. HFD).

### 3.5. Effects of HFD and CGA on Oxidative Stress and Inflammatory Cytokines

Chronic HFD feeding significantly increased oxidative stress as well as pro-inflammatory cytokine levels against control animals, indicating systemic inflammation and redox imbalance, while such effects were effectively mitigated by CGA administration (Figure 6A–E). Total antioxidant capacity (TAC) was markedly reduced in HFD-fed rats (*p* < 0.001 vs. control), reflecting impaired endogenous antioxidant defenses. Both doses of CGA significantly restored TAC levels, with the higher dose (100 mg) nearly normalizing antioxidant capacity (*p* < 0.05 and *p* < 0.001 vs. HFD, respectively). Conversely, malondialdehyde (MDA) was significantly elevated in the HFD group (*p* < 0.001 vs. control). Oral administration with CGA significantly reduced MDA levels (*p* < 0.001 and *p* < 0.0001 vs. HFD), indicating protection against oxidative injury. Interleukin-1β (IL-1β) was also significantly raised in HFD-fed rats (*p* < 0.001 vs. control). This increase was attenuated by both doses of CGA (*p* < 0.01 vs. HFD). Similarly, tumor necrosis factor-alpha (TNF-α), a central mediator of systemic inflammation, was significantly elevated under HFD conditions (*p* < 0.001 vs. control). Administration with CGA showed a difference between the low- and high-dose groups’ reduction in TNF-α levels (*p* < 0.01 and *p* < 0.001 vs. HFD). Lastly, interleukin-10 (IL-10) was significantly reduced in the HFD group (*p* < 0.001 vs. control). CGA administration significantly increased IL-10 concentrations (*p* < 0.05 vs. HFD), suggesting anti-inflammatory as well as metabolic protective effects.

### 3.6. Effects of HFD and CGA on Hypothalamic Appetite-Regulating Gene Expression

Chronic HFD feeding significantly altered hypothalamic mRNA expression of key appetite-regulating peptides compared to control animals, indicating dysregulated energy homeostasis; meanwhile, such effects were partially reversed by CGA administration (Figure 7A–D). Agrp mRNA levels were markedly upregulated in HFD-fed rats (*p* < 0.0001 vs. control), consistent with increased orexigenic signaling under obese conditions. Both doses of CGA significantly downregulated Agrp expression (*p* < 0.05 and *p* < 0.0001 vs. HFD, respectively), suggesting suppression of hunger-promoting pathways. Similarly, hypothalamic mRNA expression of NPY, another orexigenic neuropeptide, showed a significant upregulation in HFD-fed rats (*p* < 0.0001 vs. control). Treatment with CGA showed a difference between the low- and high-dose groups’ reduction in NPY mRNA levels (*p* < 0.001 and *p* < 0.0001 vs. HFD, respectively), reinforcing its role in modulating appetite regulation. In contrast, POMC, an anorexigenic gene critical for satiety signaling, was significantly downregulated in HFD-fed animals (*p* < 0.01 vs. control). Administration with 100 mg CGA upregulated POMC mRNA expression (*p* < 0.01 vs. HFD). Lastly, CARTPT, which encodes CART, a neuropeptide involved in suppressing food intake, was also significantly downregulated under HFD conditions (*p* < 0.01 vs. control). A 100 mg CGA treatment significantly upregulated CARTPT mRNA levels (*p* < 0.01 vs. HFD).

### 3.7. Effects of HFD and CGA on Abdominal WAT Inflammatory Gene Expression and miR-146a Regulation

Chronic HFD feeding significantly upregulated pro-inflammatory signaling pathways in abdominal white adipose tissue (WAT), as shown by the upregulation of mRNA expression of key inflammatory mediators, which were attenuated by CGA administration (Figure 8A–K). Such findings suggest that CGA modulates visceral adipose tissue inflammation, at least in part through regulation of the miR-146a-IRAK1-TRAF6 signaling axis. miR-146a expression was markedly downregulated in the abdominal WAT of HFD-fed rats (*p* < 0.001 vs. control), indicating impaired anti-inflammatory microRNA regulation. Both doses of CGA significantly upregulated miR-146a levels (*p* < 0.05 and *p* < 0.001 vs. HFD, respectively). Downstream targets of miR-146a, IRAK1, and TRAF6 were significantly elevated in HFD-fed rats in comparison to control rats (*p* < 0.05 and *p* < 0.001 vs. HFD, respectively). CGA treatment showed a significant reduction in both IRAK1, particularly at the higher dose (*p* < 0.05 vs. HFD), and TRAF6 (*p* < 0.05 and *p* < 0.01 vs. HFD) expression. TNF-α mRNA was significantly upregulated in HFD-fed rats compared to the control group (*p* < 0.0001 vs. control), consistent with obesity-induced inflammation. Both doses of CGA significantly reduced TNF-α expression (*p* < 0.01 and *p* < 0.001 vs. HFD, respectively). Similarly, NF-κB p65 (NFKBp65) mRNA, a central mediator of inflammatory signaling, was significantly upregulated under HFD conditions (*p* < 0.0001 vs. control). This increase was significantly suppressed by CGA administration (*p* < 0.05 and *p* < 0.001 vs. HFD, respectively). TGF-β mRNA, a pleiotropic cytokine involved in fibrosis and immune regulation, was significantly upregulated in HFD-fed rats in comparison to control animals (*p* < 0.01 vs. control). This increase was significantly reduced with only the 100 mg dose of CGA (*p* < 0.001 vs. HFD, respectively). IL-1β mRNA, an upstream mediator of inflammation, was significantly elevated in HFD-fed rats (*p* < 0.01 vs. control). Administration with 100 mg CGA led to a significant downregulation in IL-1β mRNA levels (*p* < 0.05 vs. HFD). IL-6 mRNA, a key inflammatory cytokine linked to insulin resistance, was also significantly increased in the HFD-fed rats (*p* < 0.05 vs. control). IL-8 mRNA, a chemokine involved in neutrophil recruitment and inflammation, showed a significant upregulation in HFD-fed rats (*p* < 0.0001 vs. control), and this was attenuated by CGA (*p* < 0.05 and *p* < 0.001 vs. HFD). In contrast, IL-10 mRNA, an anti-inflammatory cytokine, showed a significant downregulation in HFD-fed rats (*p* < 0.05 vs. control), indicating impaired resolution of inflammation. CGA treatment partially restored IL-10 levels at the higher dose (*p* < 0.05 vs. HFD). Lastly, IFN-γ mRNA, a marker of Th1-mediated inflammation, was significantly elevated in the HFD group (*p* < 0.0001 vs. control) and was significantly reduced following both doses of CGA (*p* < 0.001 and *p* < 0.0001 vs. HFD, respectively).

### 3.8. Effects of HFD and CGA on Apoptosis and Oxidative Stress Pathways in Abdominal White Adipose Tissue

Chronic HFD feed significantly upregulated pro-apoptotic signaling and suppressed antioxidant defenses in abdominal white adipose tissue (WAT), indicating increased cellular stress and oxidative damage; meanwhile, these effects were effectively modulated by CGA administration (Figure 9A–I), suggesting its protective role against HFD-induced adipose dysfunction. p53 mRNA, a central regulator of oxidative stress and apoptosis, was significantly elevated in HFD-fed rats compared to control animals (*p* < 0.001 vs. control), indicating heightened cellular stress. Both doses of CGA significantly reduced p53 expression (*p* < 0.01 vs. HFD). Similarly, Bax mRNA was significantly upregulated under HFD conditions (*p* < 0.001 vs. control). This increase was attenuated by CGA treatment (*p* < 0.05, and *p* < 0.001 vs. HFD, respectively). In contrast, Bcl-2 mRNA, an anti-apoptotic protein, was significantly downregulated in HFD-fed rats (*p* < 0.01 vs. control), reflecting impaired cell survival signaling. Administration with 100 mg CGA restored Bcl-2 levels (*p* < 0.001 vs. HFD). Caspase-3 mRNA, a key executioner of apoptosis, was also significantly upregulated in the HFD-fed rats (*p* < 0.001 vs. control), consistent with increased apoptotic activity in visceral adipose tissue. Treatment with CGA at both doses significantly reduced Caspase-3 expression (*p* < 0.05 and *p* < 0.01 vs. HFD, respectively). Regarding antioxidant pathways, NRF2 mRNA, a master regulator of redox homeostasis, was markedly suppressed under HFD conditions (*p* < 0.0001 vs. control). This suppression was significantly reversed by chlorogenic acid, especially at the 100 mg dose (*p* < 0.001 vs. HFD). Downstream antioxidant enzyme targets of NRF2—including heme oxygenase-1 (HO1), SOD, CAT, and GPx—were all significantly reduced in abdominal WAT of HFD-fed rat, indicating impaired ROS detoxification capacity. HO1 mRNA, which encodes heme oxygenase-1, a critical enzyme involved in cellular protection against oxidative injury, was significantly downregulated in HFD-fed rats compared to control rats (*p* < 0.05 vs. control). Administration of 100 mg CGA significantly upregulated HO1 mRNA expression (*p* < 0.05 vs. HFD, respectively). Similarly, SOD mRNA, encoding superoxide dismutase, an essential scavenger of superoxide radicals, was significantly downregulated under HFD conditions (*p* < 0.01 vs. control). Administration with CGA restored SOD levels (*p* < 0.05 and *p* < 0.0001 vs. HFD, respectively). CAT mRNA, which codes for catalase—an enzyme responsible for hydrogen peroxide detoxification—was also significantly decreased in the HFD-fed rats (*p* < 0.01 vs. control). Oral administration with CGA significantly elevated CAT mRNA expression (*p* < 0.001 and *p* < 0.0001 vs. HFD, respectively). Lastly, GPx mRNA, representing glutathione peroxidase, another key antioxidant enzyme, was significantly suppressed in HFD-fed rats (*p* < 0.01 vs. control). This suppression showed a significant downregulation by both doses of CGA (*p* < 0.05 and *p* < 0.001 vs. HFD).

## 4. Discussion

The present study demonstrates that chronic HFD feeding in Sprague Dawley rats induces significant metabolic, hormonal, hepatic, renal, oxidative stress, and inflammatory-related disturbances, consistent with the hallmarks of obesity and associated metabolic syndrome. These effects were accompanied by dysregulation of hypothalamic appetite-regulating pathways, increased visceral adiposity, systemic inflammation, adipose tissue apoptosis, and impaired antioxidant defenses. However, administration with CGA, particularly at a high dose of 100 mg/kg/day, showed a significant attenuation in these adverse effects across multiple physiological systems, suggesting its broad therapeutic potential in diet-induced metabolic dysfunction.

Chronic HFD feeding showed a significant rise in body weight, abdominal circumference, and WAT mass, confirming the development of obesity and visceral adiposity. These findings align with previous studies showing that excessive intake of dietary fats promotes energy imbalance, adipocyte hypertrophy, and insulin resistance [3]. Notably, both low (10 mg/kg) and high (100 mg/kg) levels of CGA significantly decreased body weight gain as well as visceral fat accumulation, indicating anti-obesogenic properties. While the higher CGA dose generally demonstrated more pronounced improvements in anthropometric and metabolic parameters, not all differences between the low- and high-dose groups reached statistical significance, indicating that further dose-response studies may be needed to confirm a true dose-dependent effect on energy metabolism, possibly through modulation of lipid absorption, fatty acid oxidation, and thermogenesis [16,42].

BMI, calculated as final body weight normalized to body length squared, was also elevated in HFD-fed animals, reflecting disproportionate fat accumulation relative to body size. This increase was mitigated by CGA treatment, further supporting its role in improving body composition and reducing central obesity. These findings are consistent with prior reports showing that CGA reduces adiposity and improves metabolic health in obese animal models [17]. Serum metabolic hormones, including insulin, leptin, ghrelin, and adiponectin, were significantly altered by HFD, indicating disrupted glucose homeostasis and adipokine signaling. Insulin resistance, as shown by elevated FBG and serum insulin, was effectively ameliorated by CGA, particularly at the higher dose. These findings support the notion that CGA enhances insulin sensitivity, potentially via activation of MPK, which regulates glucose uptake and lipid metabolism [19].

In this study, HFD-fed rats exhibited markedly elevated serum insulin levels and increased FBG concentrations, indicative of insulin resistance and glucose intolerance—hallmarks of metabolic syndrome and T2DM [43]. Insulin resistance arises from chronic nutrient overload, which activates inflammatory pathways including NF-κB and JNK in the liver, muscle, and adipose tissue, thereby impairing insulin signaling through serine phosphorylation of IRS-1 [3,44]. CGA treatment led to significant reductions in both serum insulin and FBG levels, suggesting improved insulin sensitivity. These findings are supported by previous studies showing that CGA improves glucose uptake in skeletal muscle and reduces hepatic gluconeogenesis via AMPK activation, a master regulator of energy homeostasis [19,42]. Moreover, CGA has been shown to inhibit intestinal glucose absorption through suppression of α-glucosidase activity, further supporting its anti-hyperglycemic effects [16].

Leptin acts on hypothalamic receptors to suppress appetite and increase thermogenesis [45]. However, in the context of chronic HFD feeding, excessive fat accumulation leads to hyperleptinemia, which paradoxically contributes to leptin resistance—a condition where central responsiveness to leptin becomes desensitized due to impaired signal transduction and increased expression of suppressors of cytokine signaling (SOCS3) [46]. In our study, HFD-fed animals displayed significantly elevated serum leptin levels, consistent with increased adiposity and leptin resistance. Notably, CGA administration effectively reduced leptin concentrations, especially at the higher dose, suggesting a regulatory effect on adipocyte function and leptin secretion. This finding is corroborated by previous reports showing that CGA can modulate adipokine secretion and improve leptin sensitivity through anti-inflammatory and antioxidant mechanisms [16,47]. Ghrelin stimulates appetite and promotes weight gain by activating orexigenic pathways in the hypothalamus [48]. Under normal physiological conditions, ghrelin rises during fasting and falls after meals, reflecting its role in meal initiation. However, in states of metabolic dysregulation such as obesity, ghrelin signaling becomes disrupted, potentially contributing to overeating and further weight gain [49]. Our results showed a significant elevation in serum ghrelin levels in HFD conditions, which may reflect compensatory hyperphagia in response to impaired satiety signals. Interestingly, only the high dose of CGA (100 mg/kg/day) was able to partially restore ghrelin levels toward control values, indicating differences between the low- and high-dose groups in modulation of appetite-regulating hormones. This observation supports the hypothesis that CGA exerts central effects on hypothalamic appetite circuits, as suggested by other studies showing CGA’s ability to pass the blood–brain barrier and influence NPY and POMC signaling [50]. Adiponectin is a pleiotropic adipokine with potent anti-inflammatory, insulin-sensitizing, and anti-atherogenic properties. It enhances insulin sensitivity by activating AMPK in liver and skeletal muscle, enhancing fatty acid oxidation, and lowering hepatic glucose production [51]. However, in obese individuals and animal models of HFD-induced obesity, adiponectin levels are typically suppressed, correlating inversely with visceral adiposity and systemic inflammation [52]. Consistent with these findings, our study showed a significant reduction in serum adiponectin in HFD-fed rats, reflecting impaired metabolic function. Remarkably, treatment with the high dose of CGA restored adiponectin levels, underscoring its capacity to reverse adipokine imbalance and promote metabolic health. This effect may be mediated through suppression of NF-κB and upregulation of PPARγ, which are known to regulate adiponectin gene expression [16,42]. It should be noted that while GAPDH was used as a reference gene for normalization of gene expression, its stability under metabolic stress conditions has been questioned in some studies. Future work may benefit from validating multiple reference genes to enhance the accuracy of mRNA quantification.

HFD feeding significantly disrupted lipid homeostasis, as evidenced by elevated total cholesterol, TAG, LDL-c, and VLDL-c, along with reduced HDL-c. These dyslipidemic effects were effectively attenuated by CGA administration. Total cholesterol and TAG levels were significantly reduced, particularly with the higher CGA dose, likely due to inhibition of lipogenic enzymes or enhanced fatty acid oxidation [42]. LDL-c and VLDL-c, markers of cardiovascular risk, were also significantly lowered by CGA, while HDL-c levels were restored, highlighting CGA’s cardioprotective potential. These results align with previous reports showing that CGA modulates lipid metabolism through activation of PPARα and suppression of SREBP-1c, leading to reduced hepatic lipid synthesis and improved lipid clearance [16,19].

Elevated serum ALT, ALP, AST, creatinine, BUN, and uric acid levels in HFD conditions indicated hepatic and renal dysfunction, consistent with previous studies linking obesity to NAFLD and kidney injury [44]. Both doses of CGA significantly reduced these biomarkers, suggesting hepatoprotective and renoprotective effects. The observed reductions in ALT and AST levels imply decreased hepatocellular damage, while lower creatinine and BUN levels suggest improved glomerular filtration and renal function. These findings align with studies showing that CGA protects against liver injury through antioxidant and anti-inflammatory mechanisms [53,54]. Furthermore, CGA’s ability to reduce uric acid suggests its broader effect in mitigating metabolic complications.

At the molecular level, HFD feeding significantly altered hypothalamic gene expression of orexigenic (Agrp, Npy) and anorexigenic (Pomc, Cartpt) neuropeptides, indicating dysregulated energy homeostasis. CGA administration partially reversed these changes, suppressing orexigenic signaling and enhancing anorexigenic pathways. These findings suggest that CGA may influence appetite regulation through both direct and indirect mechanisms. While previous studies have shown that CGA can cross the blood–brain barrier and modulate hypothalamic neuropeptides [50,55,56], our data support a role for CGA in regulating orexigenic and anorexigenic gene expression in the hypothalamus. However, we did not directly assess CNS penetration or gut hormone signaling, and future studies incorporating neuroimaging, brain microdialysis, or gut hormone profiling would be needed to confirm the central versus peripheral contributions to CGA’s anti-obesity effects. Meanwhile, our study lacks data on circulating CGA levels and its metabolites, which would help correlate systemic exposure with observed biological effects. Additionally, while CGA treatment significantly influenced appetite-regulating genes in the hypothalamus, we did not measure peripheral gut hormones such as GLP-1 or CCK beyond serum ghrelin. Recent studies suggest that polyphenols like CGA may interact with taste receptor type 2 (T2R) pathways to modulate gut hormone release, potentially contributing to appetite regulation and glucose homeostasis [57]. These receptors, traditionally associated with gustatory perception in the oral cavity, are now recognized to be expressed in various extra-oral tissues, including gastrointestinal enteroendocrine cells, where they play a pivotal role in nutrient sensing and metabolic regulation [57]. Notably, T2R activation by dietary bitter compounds—such as CGA—has been shown to stimulate the release of gut hormones, including glucagon-like peptide-1 (GLP-1) and cholecystokinin (CCK), which are involved in satiety signaling, insulin secretion, and glucose homeostasis. Importantly, recent clinical and epidemiological studies have demonstrated that genetic polymorphisms in T2R genes are significantly associated with body weight regulation, food preferences, and obesity susceptibility. As highlighted by Trius-Soler and Moreno [57], individuals carrying functional variants of TAS2R38—a key bitter taste receptor gene—exhibit distinct metabolic phenotypes. Specifically, those with the PAV/PAV haplotype (supertasters) tend to have lower BMI, reduced visceral adiposity, and decreased preference for high-fat and sweet foods compared to individuals with the non-functional AVI/AVI haplotype (non-tasters). This genetic variation influences not only taste perception but also post-ingestive metabolic responses, suggesting a direct link between T2R signaling and energy balance. Future investigations should explore these mechanisms to better understand how CGA affects central and peripheral appetite signaling. In this study, HFD-fed rats exhibited a marked reduction in total antioxidant capacity (TAC), indicating impaired redox homeostasis and increased susceptibility to oxidative damage. This observation supports the well-established link between nutrient overload and oxidative stress, wherein excess fatty acids are oxidized in mitochondria and peroxisomes, leading to excessive ROS production and diminution of antioxidant enzymes [7,13]. The observed elevation in MDA further confirms the presence of significant oxidative injury in HFD-fed animals. CGA administration effectively inverted these disturbances, restoring TAC and reducing MDA levels. These results are consistent with previous studies demonstrating CGA’s potent antioxidant properties, which include direct free radical scavenging activity as well as upregulation of endogenous antioxidant defense systems via activation of the Nrf2/Keap1 pathway [58,59]. By enhancing the expression and activity of SOD, CAT, and GPx, CGA likely mitigates oxidative damage and preserves cellular function in metabolically stressed tissues. Alongside oxidative stress, HFD feeding significantly elevated serum levels of pro-inflammatory cytokines. These cytokines are central mediators of metabolic inflammation, contributing to insulin resistance, adipocyte dysfunction, and hepatic steatosis through upregulating NF-κB and JNK signaling pathways [9,60]. TNF-α, in particular, impairs insulin signaling by promoting serine phosphorylation of IRS-1, while IL-1β exacerbates pancreatic β-cell dysfunction and contributes to the development of type 2 diabetes [61]. Both doses of CGA significantly attenuated TNF-α and IL-1β levels, suggesting a strong anti-inflammatory effect. This is supported by prior work showing that CGA suppresses LPS-induced NF-κB upregulation and cytokine release in macrophages and adipocytes [16,22]. Notably, CGA may also modulate inflammasome activity, particularly the NLRP3 inflammasome [62].

Importantly, our study found that HFD feeding significantly decreased circulating levels of IL-10. Reduced IL-10 levels have been associated with increased adipose tissue macrophage infiltration and sustained low-grade inflammation in obesity [63]. Administration with CGA led to a significant rise in IL-10 concentrations, especially at the higher dose, indicating a shift to the anti-inflammatory milieu. This finding is consistent with reports that polyphenols, including CGA, promote regulatory T cell differentiation and enhance IL-10 production, thereby counterbalancing the pro-inflammatory tone in metabolic tissues [64]. Abdominal WAT showed marked upregulation of pro-inflammatory genes, including TNF-α, IL-1β, IL-6, IL-8, IFN-γ, and TGF-β, alongside upregulation of NF-κB, indicating robust inflammation in visceral adipose tissue. CGA significantly downregulated these inflammatory mediators, particularly at the higher dose, suggesting a direct anti-inflammatory action in adipose tissue.

MicroRNA-146a (miR-146a) has emerged as a key negative regulator of inflammation, particularly in the context of metabolic disorders such as obesity. It functions by targeting critical components of the Toll-like receptor (TLR) and interleukin-1 receptor (IL-1R) signaling pathways—specifically IL-1 receptor-associated kinase 1 (IRAK1) and tumor necrosis factor receptor-associated factor 6 (TRAF6)—which are upstream activators of nuclear factor kappa B (NF-κB), a central mediator of inflammatory gene expression. Under normal physiological conditions, miR-146a serves as part of a negative feedback loop to prevent excessive NF-κB activation and cytokine production, thereby maintaining immune homeostasis [65]. In our study, we observed a significant downregulation of miR-146a in abdominal white adipose tissue (WAT) of HFD-fed rats, consistent with previous reports linking miR-146a suppression to obesity-induced inflammation [23,66,67]. Notably, treatment with chlorogenic acid (CGA), particularly at the higher dose (100 mg/kg/day), significantly restored miR-146a levels. This restoration was accompanied by a marked reduction in IRAK1 and TRAF6 expression, suggesting that CGA exerts its anti-inflammatory effects—at least in part—through modulation of the miR-146a-IRAK1-TRAF6 axis. Moreover, CGA modulated miR-146a-IRAK1-TRAF6 signaling, a key negative feedback loop in inflammation. miR-146a was markedly downregulated in HFD-fed rats, while its downstream targets, IRAK1 and TRAF6, were significantly elevated. This dysregulation suggests a failure of microRNA-mediated immune homeostasis under obese conditions, contributing to persistent inflammation. miR-146a was downregulated in HFD-fed animals, but CGA restored its expression, leading to suppression of IRAK1 and TRAF6, downstream targets of Toll-like receptor signaling. These findings suggest that CGA showed its anti-inflammatory actions, in part, through microRNA-mediated regulation of inflammatory pathways [24,56]. Future investigations should explore the broader miRNA landscape, including miR-155 and miR-21, which are also implicated in metabolic inflammation. Additionally, while mRNA expression changes were robustly demonstrated, future studies employing protein-level validation (e.g., western blotting) of IRAK1, TRAF6, and NF-κB p65 would further substantiate the mechanistic pathways proposed in this study.

In addition to inflammation, our data revealed that HFD feeding activates pro-apoptotic signaling in abdominal WAT, as evidenced by elevated expression of p53, Bax, and Caspase-3, alongside reduced expression of the anti-apoptotic protein Bcl-2. These findings are consistent with the growing body of literature linking nutrient overload and oxidative stress to mitochondrial dysfunction and programmed cell death in adipocytes [7,44]. Apoptosis of adipocytes contributes to adipose tissue dysfunction, fibrosis, and local inflammation [68,69].

Interestingly, CGA significantly suppressed apoptotic signaling, particularly at the higher dose, by reducing p53 and Bax expression while restoring Bcl-2 levels. These anti-apoptotic effects may help preserve adipocyte integrity and function, potentially preventing the onset of adipose tissue remodeling and fibrosis associated with obesity. While our findings primarily highlight the role of miR-146a in suppressing IRAK1 and TRAF6 expression, thereby reducing NF-κB-mediated inflammation, it is important to consider miR-146a’s broader regulatory functions. Recent studies have linked miR-146a to the modulation of apoptotic pathways, particularly through interactions with components of the p53-Bax/Bcl-2 axis [70,71]. In our study, we observed upregulation of miR-146a alongside increased Bcl-2 expression and reduced Caspase-3 and Bax levels in abdominal white adipose tissue (WAT). This apparent contradiction—where miR-146a, typically associated with pro-apoptotic signaling in certain contexts, correlates with enhanced cell survival in WAT—warrants further investigation. One possible explanation is that miR-146a may exert tissue-specific effects or interact with compensatory mechanisms under conditions of metabolic stress. Future studies exploring miR-146a’s functional role in apoptosis-related pathways will help clarify its complex regulatory effects in different tissues.

Furthermore, HFD-fed animals exhibited marked suppression of NRF2, the master regulator of antioxidant gene expression, along with reduced levels of downstream antioxidant enzymes. This indicates impaired redox defense mechanisms in WAT, leaving adipocytes vulnerable to ROS-induced injury. Indeed, elevated MDA levels in the serum support the presence of systemic oxidative damage in HFD-fed animals.

Chlorogenic acid, particularly at the 100 mg/kg dose, effectively restored NRF2 expression and enhanced antioxidant enzyme activity, suggesting that its protective effects are mediated, at least in part, through activation of the NRF2/Keap1 pathway. By enhancing endogenous antioxidant defenses, CGA likely reduces ROS burden and prevents oxidative damage in metabolically active tissues, thus preserving cellular function and metabolic homeostasis [72,73]. Several limitations must be acknowledged. First, the duration of CGA treatment was relatively short (4 weeks), and longer-term studies are needed to evaluate sustained effects and safety. Second, only male rats were used, limiting generalizability to female populations. Third, pharmacokinetic and bioavailability data for CGA in this rat model were not included, and these data are crucial for interpreting dose-response relationships and guiding clinical translation. Based on allometric scaling calculations, the high CGA dose used in this study (100 mg/kg in rats) translates to approximately 16 mg/kg in humans, which would correspond to a daily intake of around 800–1000 mg for an average adult. However, it should be noted that the bioavailability of CGA varies significantly across species due to differences in gut microbiota composition and metabolic processing. Furthermore, dietary intake of CGA from natural sources (e.g., coffee, fruits, vegetables) may not reach these pharmacological concentrations consistently, suggesting that administration might be necessary to achieve therapeutic effects. Future studies should explore formulation strategies to enhance CGA bioavailability and assess long-term safety in human populations. In addition, the inclusion of a positive control group, such as a known anti-obesity or insulin-sensitizing agent (e.g., metformin) would allow a direct comparison of CGA’s efficacy. Future studies incorporating such controls are warranted to better contextualize the therapeutic potential of CGA relative to existing interventions. Another limitation of the present study is the exclusive reliance on mRNA expression data for inflammatory and apoptotic markers. Protein-level validation and functional assays would strengthen the interpretation of these findings and are recommended for future investigations.

## 5. Conclusions

In conclusion, this study showed compelling potential for the therapeutic efficacy of CGA in mitigating HFD-induced metabolic dysfunction. CGA administration significantly improved body weight, adiposity, glucose as well as lipid metabolism, hepatic and renal function, inflammation, oxidative stress, and central appetite regulation. Its protective effects appear to be mediated through multiple mechanisms, including modulation of hypothalamic neuropeptide expression, suppression of NF-κB and TLR signaling via miR-146a, enhancement of antioxidant defenses via NRF2 activation, and inhibition of adipose tissue apoptosis. Given the global epidemic of obesity and related metabolic disorders, natural compounds like CGA offer promising avenues for preventive and therapeutic interventions. Future studies should explore the long-term safety, bioavailability, and clinical translation of CGA in human populations. Additionally, investigations into combination therapies, microbiome interactions, and sex-based differences will further elucidate the full potential of CGA in metabolic health.

## Figures and Tables

**Figure 1 biomolecules-15-01086-f001:**
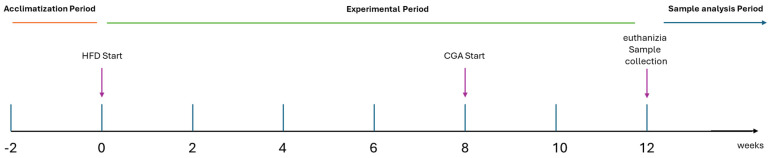
Schematic representation of the experimental timeline.

**Figure 2 biomolecules-15-01086-f002:**
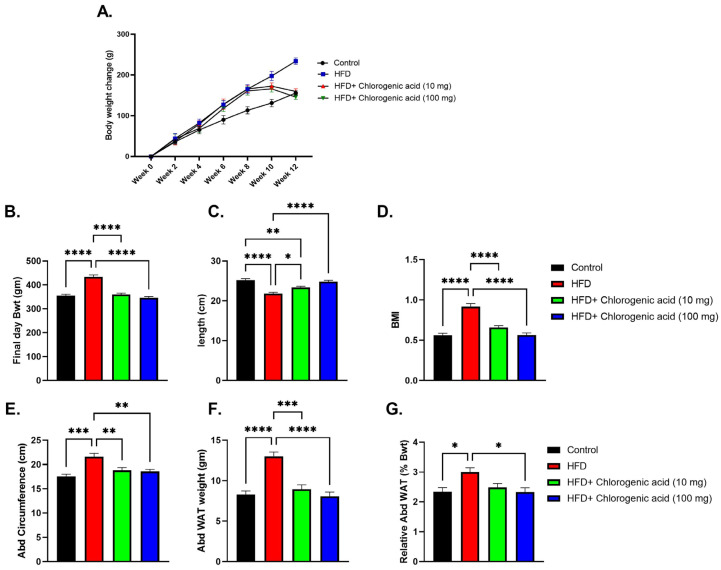
Effects of HFD and CGA oral administration on anthropometric and metabolic parameters (**A**–**F**): (**A**) body weight change (g), (**B**) final body weight (g), (**C**) body length (cm), (**D**) body mass index (BMI), (**E**) abdominal circumference (cm), (**F**) abdominal WAT weight (mg), and (**G**) relative abdominal WAT weight (mg/g body weight). Data are shown as mean ± SEM (N = eight). Statistical significance: *, **, ***, **** indicate *p* < 0.05, 0.01, 0.001, 0.0001.

**Figure 3 biomolecules-15-01086-f003:**
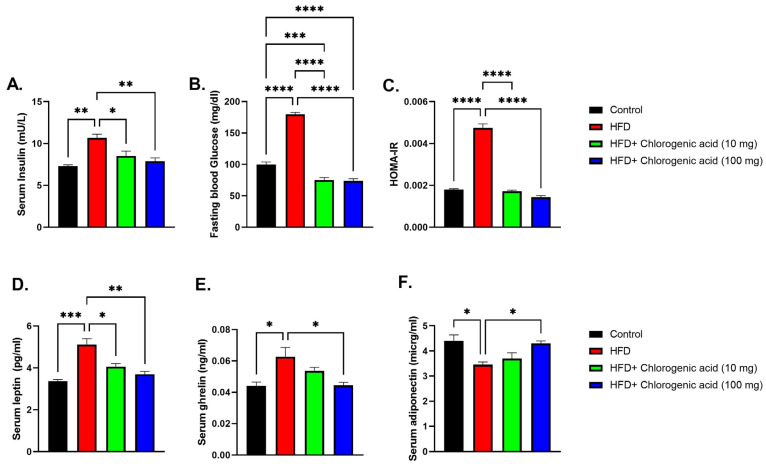
Effects of HFD and CGA oral administration on metabolic hormone levels (**A**–**E**): (**A**) serum insulin (mU/L), (**B**) FBG (mg/dL), (**C**) HOMA-IR, (**D**) serum leptin (pg/mL), (**E**) serum ghrelin (ng/mL), (**F**) serum adiponectin (μg/mL). Data are shown as mean ± SEM (N = eight). Statistical significance: *, **, ***, **** indicate *p* < 0.05, 0.01, 0.001, 0.0001.

**Figure 4 biomolecules-15-01086-f004:**
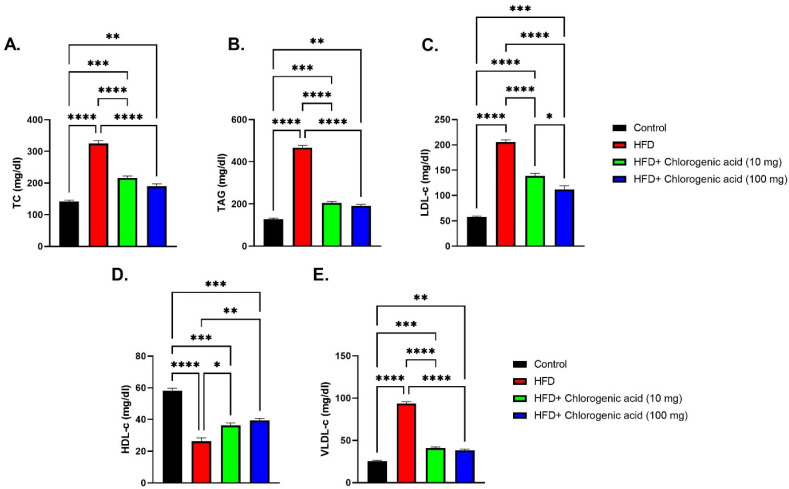
Effects of HFD and CGA oral administration on lipid profile parameters (**A**–**E**): (**A**) TC (mg/dL), (**B**) TAG (mg/dL), (**C**) LDL-c (mg/dL), (**D**) HDL-c (mg/dL), (**E**) VLDL-c (mg/dL). Data are shown as mean ± SEM (N = eight). Statistical significance: *, **, ***, **** indicate *p* < 0.05, 0.01, 0.001, 0.0001.

**Figure 5 biomolecules-15-01086-f005:**
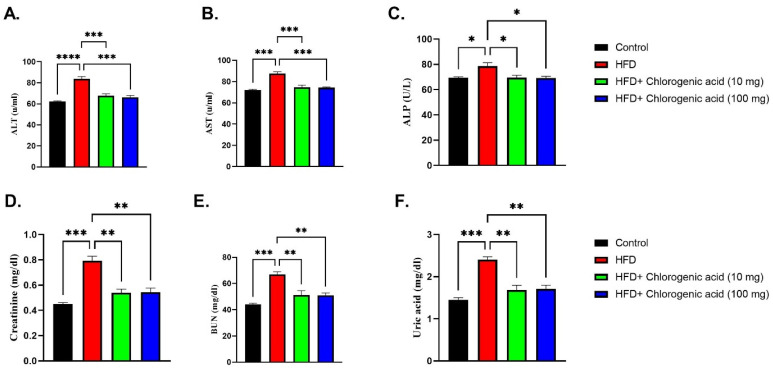
Effects of HFD and CGA oral administration on hepatic and renal function markers (**A**–**F**): (**A**) ALT (U/mL), (**B**) AST (U/mL), (**C**) ALP (U/L), (**D**) creatinine (mg/dL), (**E**) BUN (mg/dL), (**F**) uric acid (mg/dL). Data are shown as mean ± SEM (N = eight). Statistical significance: *, **, ***, **** indicate *p* < 0.05, 0.01, 0.001, 0.0001.

**Figure 6 biomolecules-15-01086-f006:**
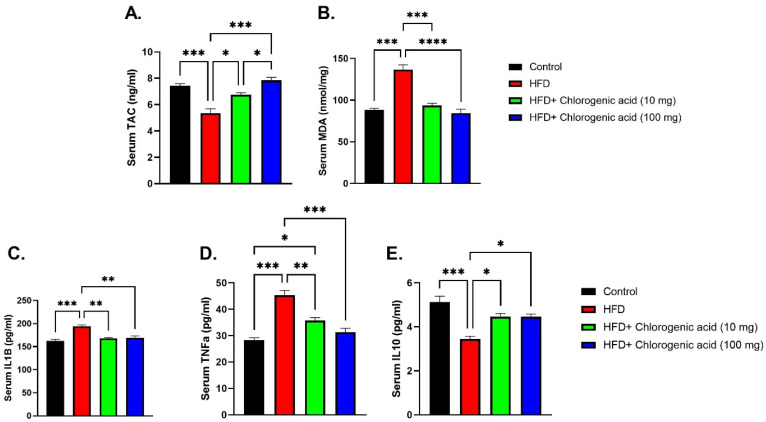
Effects of HFD and CGA oral administration on oxidative stress and inflammatory markers (**A**–**E**): (**A**) TAC (ng/mL), (**B**) MDA (nmol/mg), (**C**) IL-1β (pg/mL), (**D**) TNF-α (pg/mL), (**E**) IL-10 (pg/mL). Data are shown as mean ± SEM (N = eight). Statistical significance: *, **, ***, **** indicate *p* < 0.05, 0.01, 0.001, 0.0001.

**Figure 7 biomolecules-15-01086-f007:**
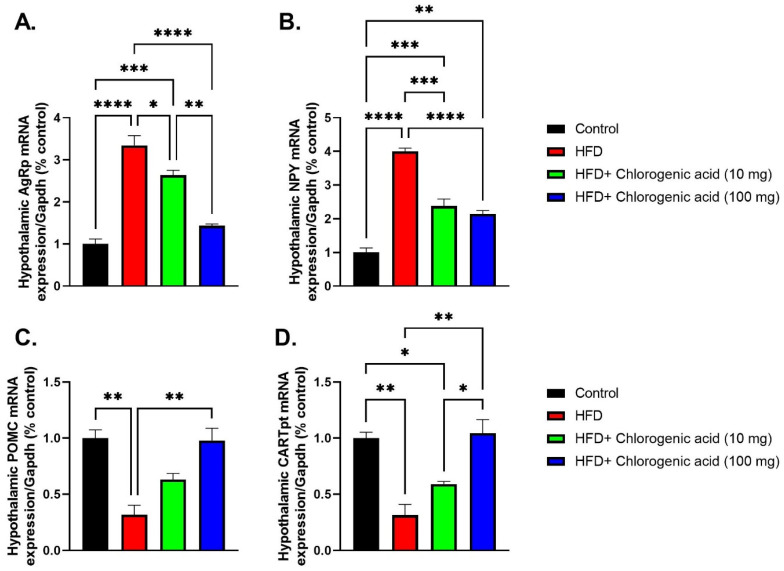
Effects of HFD and CGA oral administration on hypothalamic mRNA expression of appetite-regulating genes (**A**–**D**): (**A**) Agrp mRNA, (**B**) NPY mRNA, (**C**) POMC mRNA, (**D**) CARTPT mRNA. Data are shown as mean ± SEM (N = eight). Statistical significance: *, **, ***, **** indicate *p* < 0.05, 0.01, 0.001, 0.0001.

**Figure 8 biomolecules-15-01086-f008:**
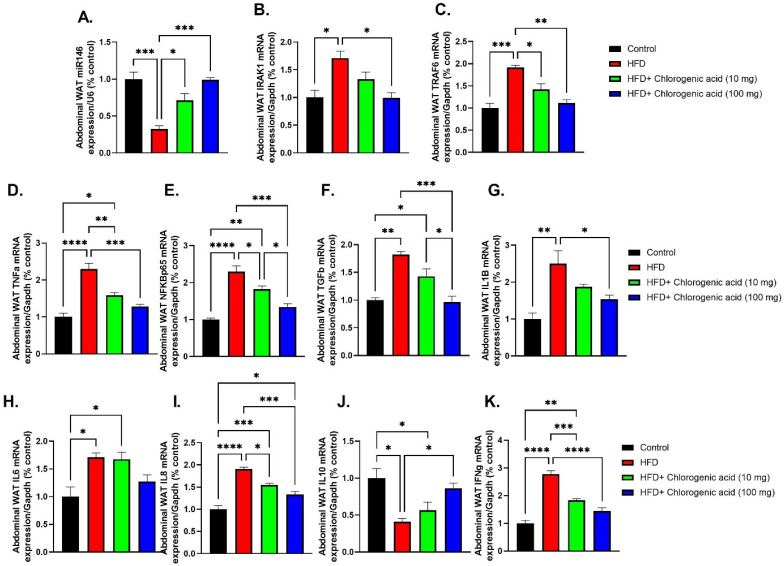
Effects of HFD and CGA oral administration on abdominal WAT inflammatory gene expression and miR-146a regulation (**A**–**K**): (**A**) miR-146a expression (% U6), (**B**) IRAK1 mRNA, (**C**) TRAF6 mRNA, (**D**) TNF-α mRNA, (**E**) NFKBp65 mRNA, (**F**) TGF-β mRNA, (**G**) IL-1β mRNA, (**H**) IL-6 mRNA, (**I**) IL-8 mRNA, (**J**) IL-10 mRNA, (**K**) IFN-γ mRNA. Data are shown as mean ± SEM (N = eight). Statistical significance: *, **, ***, **** indicate *p* < 0.05, 0.01, 0.001, 0.0001.

**Figure 9 biomolecules-15-01086-f009:**
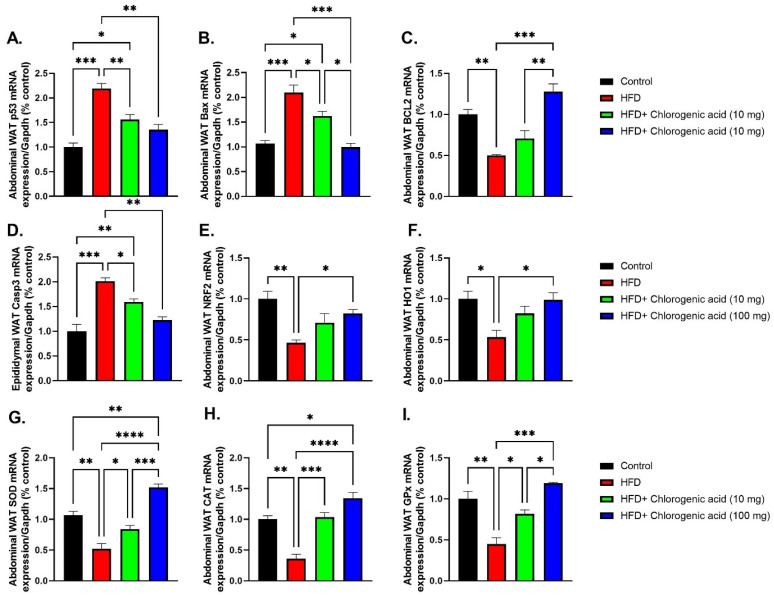
Effects of HFD and CGA oral administration on abdominal WAT mRNA expression of apoptosis- and oxidative-stress-related genes (**A**–**I**): (**A**) p53 mRNA, (**B**) Bax mRNA, (**C**) Bcl-2 mRNA, (**D**) Caspase-3 mRNA, (**E**) NRF2 mRNA, (**F**) HO1 mRNA, (**G**) SOD mRNA, (**H**) CAT mRNA, (**I**) GPx mRNA. Data are shown as mean ± SEM (N = eight). Statistical significance: *, **, ***, **** indicate *p* < 0.05, 0.01, 0.001, 0.0001.

**Table 1 biomolecules-15-01086-t001:** Composition of experimental custom-formulated high-fat diet.

Ingredient	HFD (g/kg Diet (kcal/kg))
Casein	200 (716)
L-Cystine	3 (12)
Sucrose	100 (400)
Cornstarch	173.1 (623)
Dyetrose	58 (220.4)
Lard	350 (3150)
Cellulose	50 (0)
Mineral Mix (#210025)	49.3 (43.4)
Vitamin Mix (#310025)	14.1 (54.6)
Choline Bitartrate	2.5 (0)
Total	1000 (5219.4)

**Table 2 biomolecules-15-01086-t002:** Experimental groups and different treatments.

Group	Diet	Treatment Duration	Chlorogenic Acid Dose
Control Group	Standard chow diet (10% kcal from fat)	12 weeks	Vehicle only (distilled water, orally, last 4 weeks)
HFD Group	High-fat diet (60% kcal from fat)	12 weeks	Vehicle only (distilled water, orally, last 4 weeks)
HFD + Low-Dose CGA Group	High-fat diet (60% kcal from fat)	First 8 weeks: HFD only Last 4 weeks: HFD + CGA	10 mg/kg/day, oral gavage
HFD + High-Dose CGA Group	High-fat diet (60% kcal from fat)	First 8 weeks: HFD only Last 4 weeks: HFD + CGA	100 mg/kg/day, oral gavage

**Table 3 biomolecules-15-01086-t003:** Primer sequences used in RT-PCR analysis.

Gene	Forward Primer (5′-3′)	Reverse Primer (5′-3′)	Accession Number
AGRP	AAGCTTTGGCAGAGGTGCTA	GACTCGTGCAGCCTTACACA	NM_033650.1
NPY	TACTCCGCTCTGCGACACTA	TGTCTCAGGGCTGGATCTCT	NM_012614.2
CART-pt	CCCTACTGCTGCTGCTACCT	CACGGCAGAGTAGATGTCCA	NM_017110.1
POMC	GCTTCATGACCTCCGAGAAG	TCTTGATGATGGCGTTCTTG	NM_139326.3
IRAK1	GCTGTGGACACCGAT	GCTACACCCATCCACA	NM_001127555.1
TRAF6	CAGTCCCCTGCACATT	GAGGAGGCATCGCAT	M_001107754.2
TNF-α	AGGGTCTGGGCCATAGAAC	CCACCACGCTCTTCTGTCTAC	NM_012675.3
NF-κB	CAGGACCAGGAACAGTTCGAA	CCAGGTTCTGGAAGCTATGGAT	NM_199267.2
TGFβ1	CTGAACCAAGGAGACGGAAT	GGTTCATGTCATGGATGGTG	NM_021578.2
IL1β	CACCTCTCAAGCAGAGCACAGA	ACGGGTTCCATGGTGAAGTC	NM_031512.2
IL6	ATATGTTCTCAGGGAGATCTTGGAA	GTGCATCATCGCTGTTCATACA	NM_012589.2
IL8	CATTAATATTTAACGATGTGGATGCGTTTCA	GCCTACCATCTTTAAACTGCACAAT	NM_030845.1
IL10	GTAGAAGTGATGCCCCAGGC	AGAAATCGATGACAGCGTCG	NM_012854.2
IFNg	GTGAACAACCCACAGATCCA	GAATCAGCACCGACTCCTTT	NM_138880.3
P53	CATGAGCGTTGCTCTGATGGT	GATTTCCTTCCACCCGGATAA	NM_030989.3
Bax	CGAATTGGCGATGAACTGGA	CAAACATGTCAGCTGCCACAC	NM_017059.2
Bcl-2	GACTGAGTACCTGAACCGGCATC	CTGAGCAGCGTCTTCAGAGACA	NM_016993.1
Casp-3	GAGACAGACAGTGGAACTGACGATG	GGCGCAAAGTGACTGGATGA	NM_012922.2
NRF2	CACATCCAGACAGACACCAGT	CTACAAATGGGAATGTCTCTGC	NM_031789
HO1	GTAAATGCAGTGTTGGCCCC	ATGTGCCAGGCATCTCCTTC	NM_012580.2
CAT	TCCATCCTTTATCCATAGCC	TTAACCAGCTTGAAGGTGTG	NM_012520.2
SOD	TGTGATCTCACTCTCAGGAG	CTCAGACCACATAGGGAATG	NM_017050.1
GPx	GCGTCCCTCTGAGGCACCAC	AAGTTGGGCTCGAACCCACC	NM_030826.4
Gapdh	GTGCCAGCCTCGTCTCATAG	CGTTGATGGCAACAATGTCCA	NM_017008.4
U6	GCTCGCTTCGGCAGCACA	GAGGTATTCGCACCAGAGGA	
miR146a	GTTTGGTGAGAACTGAATTCCA	GTGCAGGGTCCGAGGT	
U6 stem-Loop primer	AACGCTTCACGAATTTGCGTG		
miR14 stem-Loop primer	GTTGGCTCTGGTGCAGGGTCCGAGGTATTCGCACCAGAGCCAACAACCCA		

## Data Availability

The original contributions presented in this study are included in the article. Further inquiries can be directed to the corresponding author.

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
