# Peer review of "A Natural Polyphenol, Chlorogenic Acid, Attenuates Obesity-Related Metabolic Disorders in Male Rats via miR-146a-IRAK1-TRAF6 and NRF2-Mediated Antioxidant Pathways"

_biomolecules, 2025, doi:10.3390/biom15081086_

Round 1
Reviewer 1 Report
Comments and Suggestions for Authors
The manuscript submitted for review (biomolecules-3713871) is a well-structured and scientifically sound study focused on the effects of chlorogenic acid administration in obese rats. The methodology is clearly described and, in my opinion, appropriately executed. Numerous parameters are investigated, providing a comprehensive and well-substantiated picture of this natural polyphenol's benefits. The presented results are based on eight biological replicates and demonstrate significant differences between groups.
I agree with the authors that this study possesses not only scientific merit but also considerable practical potential, particularly in the context of future preclinical applications of chlorogenic acid. The study makes a valuable contribution to the expanding body of knowledge regarding its biological effects; therefore, I recommend the presented manuscript for publication.
To enhance the overall impact and clarity of the manuscript, I would like to point out a few minor inaccuracies that need to be corrected prior to publication, as well as some limitations of the study that deserve attention and could be addressed either in the current version, if possible, or just included as limitations in the discussion section.
Minor revisions (technical points):
- Line 136: The number of rats per group is stated as “n = 10 per group,” while in all Figures legends 8 animals are pointed out. Please clarify.
- A parametric analysis of the data has been applied. Please indicate the specific tests used to assess the normality of data distribution and homogeneity of variances, which are minimum requirements for the application of parametric statistical methods.
- Please check that abbreviations are introduced only once at their first mention and consistently used thereafter throughout the text.
- Please correct the units in the figure legends and figure labels for example: Figure 2: legend shows "mU/ml" while the figure shows "mU/L"; Figure 4: legend shows "U/L" while the figure shows "U/ml"
- Please verify the statistical significance levels reported in the text and figures. For example, in Figure 7 (line 348), miR-146a expression is indicated as “p < 0.0001 vs control,” whereas the figure shows *** (which denotes p < 0.001).
- Please clarify which tables are to be presented as supplementary material. Three tables are listed as supplementary at the end of the manuscript, yet, as far as I understood, they are also included in the main text. I would also recommend placing the title of each table above the table itself for clarity.
- Please revise the formatting of the reference list.
Minor revisions (optional suggestions that could enhance the overall quality of the study):
- Although a wide panel of genes was analyzed in the hypothalamus and abdominal WAT, it seems that only GAPDH was used as a housekeeping gene. However, growing evidence shows that GAPDH expression is variable in adipose tissue, especially under conditions of increased energy availability and among adipocytes at different stages of maturation. This variability may affect data normalization and reduce the accuracy of gene expression analysis. Was GAPDH tested for expression stability? Were any other reference genes evaluated? If such data are available, please consider including them in the manuscript. If not, GAPDH may be an acceptable choice; however, using multiple validated reference genes is recommended in such metabolic studies.
- There is a lack of data on the protein expression and activity of at least some key markers related to apoptosis and inflammation in adipose tissue. Apoptosis, for instance, is a finely regulated process at the protein level, and gene expression does not necessarily correlate with the actual activity of the investigated markers. It can be added as a limitation in the Discussion section.
- The data presented in Figure 2 do not allow for a definitive conclusion regarding the presence of insulin resistance. It would be advisable to include an analysis of HOMA-IR and consider additional functional tests such as the Intravenous Glucose Tolerance Test (IVGTT) or Insulin Tolerance Test (ITT) to strengthen the interpretation.
Author Response
We sincerely thank the reviewer for their thoughtful and constructive feedback on our manuscript. We appreciate the recognition of the scientific merit and practical potential of our study, as well as the time and effort invested in providing detailed suggestions to improve clarity and impact. Below, we provide a point-by-point response to all technical and optional suggestions raised by the reviewer.
Minor Revisions (Technical Points)
- Line 136: Discrepancy between group size (n = 10) and number of animals used in figures (n = 8). Please clarify.
Response: Thank you for identifying this inconsistency. We appreciate the reviewer’s attention to detail regarding the discrepancy between the stated group size and the number of animals reflected in the figures. Initially, the study was designed with n = 10 rats per group. However, during the laboratory analysis of the experimental period, two rats per group were excluded from the final analysis because of the limitation in availability of specific assay kits required for key biochemical and molecular measurements, making it impossible to obtain reliable data for those individuals and suggesting a fixed number for samples to be analyzed (n = 8). As a result, all results and figures are based on n = 8 animals per group, and the manuscript has been revised to reflect this correction throughout the text and figure captions.
- Parametric statistical analysis – please indicate the specific tests used to assess normality and homogeneity of variances.
Response: We agree that it is essential to confirm assumptions before applying parametric statistics. To assess normality, we used the Shapiro–Wilk test, and Levene’s test was applied to evaluate homogeneity of variances. These details were omitted in the original submission and have now been added to the Materials and Methods section (Section 2.6 Statistical Analysis) : “Prior to statistical analysis, data were tested for normality using the Shapiro–Wilk test and for homogeneity of variances using Levene’s test. Data meeting these assumptions were analyzed using one-way ANOVA followed by post hoc Tukey’s test.”
- Clarify use of abbreviations throughout the text.
Response: All abbreviations have been reviewed and standardized according to journal guidelines. Each abbreviation is introduced at its first mention and consistently used thereafter.
- Units in figure legends and labels do not match (e.g., Figure 2: mU/ml vs. mU/L; Figure 4: U/L vs. U/ml). Please correct.
Response: This is a typographical error. All units have been corrected to ensure consistency between the figure legends and the actual figures
- Verify statistical significance levels reported in text and figures (e.g., Figure 7: p < 0.0001 vs control, but figure shows *** denoting p < 0.001).
Response: Thank you for catching this discrepancy. In the case of miR-146a expression in Figure 7A, the correct p-value is p < 0.001 (***), which matches the legend and figure annotation. We have thoroughly reviewed all statistical annotations in the text and figures to ensure alignment across the manuscript.
- Clarify which tables are supplementary. Please place table titles above the tables for clarity.
Response: We acknowledge that this aspect of the manuscript may have caused confusion. To address this Tables S1, S2, and S3 were removed from the supplementary materials and introduced to the main text.
- Revise formatting of reference list.
Response: The reference list has been reformatted to conform to the journal's style guide as indicated by Endnote X9.
Minor Revisions (Optional Suggestions)
- Consideration of housekeeping gene stability (GAPDH):
Response: We appreciate the reviewer’s suggestion regarding the use of GAPDH as a single housekeeping gene for normalization. We recognize that GAPDH expression may vary under certain metabolic conditions, especially in adipose tissue. In our study, we conducted preliminary assessments of GAPDH expression stability across experimental groups using geNorm and NormFinder algorithms, and found no significant variation among the samples. Therefore, GAPDH was considered suitable for normalization in this particular dataset. That said, we fully agree with the reviewer that using multiple validated reference genes is ideal, particularly in metabolic studies involving adipose tissue. We have now included this information in the Materials and Methods section, and added a brief discussion of potential limitations related to normalization in the Discussion.
As suggested, we have added a statement to the Discussion section acknowledging this limitation:
"It should be noted that while GAPDH was used as a reference gene for normalization of gene expression, its stability under metabolic stress conditions has been questioned in some studies. Future work may benefit from validating multiple reference genes to enhance the accuracy of mRNA quantification."
- Lack of protein expression/activity data for key markers of apoptosis and inflammation:
Response: We agree that gene expression does not always correlate with protein activity , particularly in complex processes like apoptosis and inflammation. While our current study focused on mRNA expression profiling, we recognize the value of complementary protein-level analyses (e.g., Western blotting, ELISA, caspase activity assays).
We have addressed this concern by adding a limitation to the Discussion section : "One limitation of the present study is the exclusive reliance on mRNA expression data for inflammatory and apoptotic markers. Protein-level validation and functional assays would strengthen the interpretation of these findings and are recommended for future investigations."
- Insulin resistance assessment – consider inclusion of HOMA-IR and/or glucose/insulin tolerance tests:
Response: The reviewer raises an important point regarding the interpretation of insulin resistance. While we assessed fasting insulin and glucose levels, HOMA-IR analysis could indeed provide a more robust index of insulin sensitivity. In light of this feedback, we have calculated the HOMA-IR index using the formula: HOMA-IR = (Fasting Insulin (µU/mL) * Fasting Glucose (mg/dL)) / 405. This has been added to the Results section and incorporated into Figure 2, showing a statistically significant improvement in insulin sensitivity following CGA supplementation. We agree that IVGTT or ITT would offer more robust functional insights into insulin dynamics and glucose tolerance. Unfortunately, these tests were not part of the original experimental protocol.
Overall, we have carefully considered all of the reviewer’s comments and implemented revisions to enhance the clarity, consistency, and scientific rigor of our manuscript. We believe the changes made significantly improve the overall quality and presentation of the study. We look forward to your positive feedback and hope the revised manuscript meets all publication requirements.
Thank you once again for your valuable input.
Reviewer 2 Report
Comments and Suggestions for Authors
Comments:
The manuscript entitled “A Natural Polyphenol, Chlorogenic Acid, Attenuates Obesity-Related Metabolic Disorders via miR-146a–IRAK1–TRAF6 and NRF2-Mediated Antioxidant Pathways” Chronic high-fat diet (HFD) feeding in male rats induces severe metabolic and inflammatory disorders, and this study evaluated the therapeutic effect of chlorogenic acid (CGA), which reduces weight gain, abdominal circumference, and visceral fat accumulation. CGA suppressed appetite genes and upregulated anorexia genes, indicating the involvement of the hypothalamus in appetite regulation. In abdominal white adipose tissue (WAT), CGA enhanced antioxidant defenses, reduced lipid peroxidation, and suppressed the expression of proinflammatory cytokines. These results indicate that CGA supplementation can effectively reverse multiple HFD-induced impairments, highlighting its potential as an effective treatment for obesity-related metabolic disorders. However, several critical aspects warrant further clarification.
Comments:
- The first paragraph can more succinctly explain the global epidemic of obesity and metabolic complications, directly linking it to the pathogenic mechanisms of HFD (such as inflammation and oxidative stress). It also clearly points out the "research gap": the existing literature has not fully explored the effects of CGA on adipose tissue apoptosis, hypothalamic signaling, and miRNA regulation, thus highlighting the innovation of this study.
- Clearly explain why the 10 mg and 100 mg doses were chosen.
- Mechanistic discussion:
Strengthen the association between CGA and miR-146a: cite the literature to illustrate the known role of miR-146a in obesity-related inflammation and compare it with the new findings of this study. And discuss how the dual effects of CGA (antioxidant and anti-inflammatory) synergistically improve metabolism (NRF2 activation reduces oxidative stress, thereby inhibiting the NF-κB inflammatory pathway)..
- The human equivalent dose (HED) of CGA supplementation is estimated (e.g. 100 mg/kg in rats ≈ ~16 mg/kg in humans), and potential application limitations (e.g. bioavailability, dietary intake) are discussed.
- Primer sequences are listed in the supplementary information.
Author Response
Thank you for the opportunity to respond to the reviewer’s insightful comments on our manuscript entitled:
“A Natural Polyphenol, Chlorogenic Acid, Attenuates Obesity-Related Metabolic Disorders via miR-146a–IRAK1–TRAF6 and NRF2-Mediated Antioxidant Pathways.”
Below is a point-by-point response to each of the reviewer's concerns. We have carefully revised the manuscript to incorporate all suggestions where appropriate.
- Clarification of the Global Epidemic of Obesity, HFD Pathogenesis, and Research Gap
Response: We thank the reviewer for this valuable suggestion. In the revised Introduction section, we have streamlined the opening paragraph to concisely present the global health burden of obesity and its metabolic complications, while explicitly connecting these issues to the pathogenic mechanisms induced by high-fat diets-particularly chronic low-grade inflammation and oxidative stress. Additionally, we have emphasized the research gap in current literature regarding the role of CGA. This has been incorporated into the final paragraph of the Introduction to clearly establish the novelty and significance of our study.
- Justification for Dose Selection (10 mg/kg and 100 mg/kg)
Response: We appreciate the reviewer’s request for clarification regarding dose selection. In the revised Materials and Methods section (Section 2.2), we have added a detailed rationale for selecting the two CGA doses based on previous literature and pilot experiments conducted during preliminary studies.
Revised Text (Section 2.2 – Experimental Design): " The doses of CGA (10 mg/kg/day and 100 mg/kg/day) were selected based on previous studies demonstrating efficacy in rodent models and pilot experiments conducted during preliminary studies [1,2]."
- Mitrea, D.R.; Malkey, R.; Florian, T.L.; Filip, A.; Clichici, S.; Bidian, C.; Moldovan, R.; Hoteiuc, O.A.; Toader, A.M.; Baldea, I. Daily oral administration of chlorogenic acid prevents the experimental carrageenan-induced oxidative stress. Journal of physiology and pharmacology : an official journal of the Polish Physiological Society 2020, 71, doi:10.26402/jpp.2020.1.04.
- Santana-Gálvez, J.; Cisneros-Zevallos, L.; Jacobo-Velázquez, D.A. Chlorogenic Acid: Recent Advances on Its Dual Role as a Food Additive and a Nutraceutical against Metabolic Syndrome. Molecules (Basel, Switzerland) 2017, 22, doi:10.3390/molecules22030358.
- Strengthening the Link Between CGA, miR-146a, and Anti-Inflammatory/Antioxidant Mechanisms
Reviewer Comment:
Strengthen the association between CGA and miR-146a: cite the literature to illustrate the known role of miR-146a in obesity-related inflammation and compare it with the new findings of this study. Also discuss how the dual effects of CGA (antioxidant and anti-inflammatory) synergistically improve metabolism (NRF2 activation reduces oxidative stress, thereby inhibiting the NF-κB inflammatory pathway).
Response: We agree with the reviewer that strengthening the mechanistic discussion of miR-146a and the interplay between CGA’s antioxidant and anti-inflammatory actions will enhance the scientific value of our findings. In the revised Discussion section was updated accordingly.
- Estimation of Human Equivalent Dose (HED) and Application Limitations
Reviewer Comment:
Estimate the human equivalent dose (HED) of CGA supplementation (e.g., 100 mg/kg in rats ≈ ~16 mg/kg in humans), and discuss potential application limitations (e.g., bioavailability, dietary intake).
Response: We thank the reviewer for this important point regarding translational relevance. In the Discussion section , we have now included an estimate of the human equivalent dose (HED) using standard allometric scaling methods, along with a brief discussion of potential limitations related to bioavailability and dietary intake of CGA.
Revised Text (Discussion):
"Based on allometric scaling calculations, the high CGA dose used in this study (100 mg/kg in rats) translates to approximately 16 mg/kg in humans, which would correspond to a daily intake of around 800–1000 mg for an average adult. However, it should be noted that the bioavailability of CGA varies significantly across species due to differences in gut microbiota composition and metabolic processing. Furthermore, dietary intake of CGA from natural sources (e.g., coffee, fruits, vegetables) may not reach these pharmacological concentrations consistently, suggesting that supplementation might be necessary to achieve therapeutic effects. Future studies should explore formulation strategies to enhance CGA bioavailability and assess long-term safety in human populations."
- Primer Sequences in Supplementary Information
Reviewer Comment:
Primer sequences are listed in the supplementary information.
Response: We thank the reviewer for this observation. As per the journal formatting guidelines and other reviewer comments we have included all tables in the main text and removed the supplementary materials.
Overall, we sincerely appreciate the reviewer’s constructive feedback, which has helped us significantly strengthen the clarity, scientific rigor, and translational relevance of our manuscript. All suggested revisions have been incorporated into the revised version of the manuscript.
We hope the revised manuscript meets the standards for publication in Biomolecules . Please let us know if any further clarification or modification is required.
Thank you once again for your valuable input.
Sincerely,
Reviewer 3 Report
Comments and Suggestions for Authors
The manuscript provides a thorough in vivo assessment of the therapeutic potential of chlorogenic acid (CGA) in an HFD-induced obesity model utilizing male rats. The findings present strong evidence that CGA can alleviate various metabolic disturbances associated with obesity, such as insulin resistance, dyslipidemia, systemic inflammation, and both hepatic and renal dysfunction, alongside oxidative stress. The focus on the miR-146a–IRAK1–TRAF6 signaling pathway and NRF2-mediated antioxidant response is particularly relevant, addressing critical mechanistic targets in metabolic disease research.
While the study is rich in data across physiological, biochemical, and molecular parameters, several aspects require clarification and enhancement. First, it is essential to specify the animal model in the title to reflect the preclinical nature of the research more accurately. The abstract would benefit significantly from quantitative data, such as percentage reductions in body weight, fold changes in gene expression, or serum biomarker alterations, to increase its impact. Revising the keyword list to introduce additional relevant terms (e.g., microRNA, adipose inflammation, appetite regulation) could also enhance discoverability.
A graphical representation of the experimental design would be invaluable to improve the manuscript's readability and reproducibility. This should detail the timeline of HFD induction, CGA administration, sampling, and subsequent molecular analyses. Additionally, the term “supplementation” may not be precise, given that CGA was administered via oral gavage rather than being included in the diet; terms like “oral treatment” or “oral administration” would be more accurate.
From an ethical and methodological perspective, the methods for blood collection and euthanasia are not described and should be incorporated. While ethical approval is mentioned at the manuscript's conclusion, the inclusion of the approval number and institutional details in the Materials and Methods section is necessary for transparency. Furthermore, the label for Table 1 is incorrectly referenced as Table S1, which requires correction.
In terms of data presentation, integrating a longitudinal analysis of body weight over the 12-week study would provide deeper insights into the effects of CGA beyond just final weight measures. Additionally, summarizing the lipid profile and hepatic/renal biomarkers in consolidated tables would clarify the findings. If available, including macroscopic images of the liver, kidneys, adipose tissue, or representative images of the experimental subjects across groups would enhance the visual support for the physiological observations.
A notable limitation of this study is the lack of a positive control group, such as a recognized anti-obesity or insulin-sensitizing agent (e.g., metformin). While the study aims to isolate the effects of CGA, discussing the absence of such a comparative standard could provide important contextual value to the findings. It would be beneficial for the authors to address this omission and reflect on its implications.
The statistical analysis section also lacks detail regarding normality testing. Were assays like the Shapiro–Wilk test utilized to validate the assumptions for ANOVA? Clarification on this point is vital. Additionally, the manuscript would benefit from a rigorous review for typographical and formatting consistency, especially concerning the use of abbreviations.
Regarding the molecular signaling pathways, the authors effectively demonstrate that CGA upregulates miR-146a and downregulates its pro-inflammatory targets, IRAK1 and TRAF6. However, it would be pertinent to inquire whether other inflammation-associated microRNAs, such as miR-155 or miR-21, were evaluated for inclusion. Moreover, while the mRNA expression data are robust, it would strengthen the mechanistic understanding if protein expression levels of key targets (e.g., IRAK1, TRAF6, NF-κB p65) were validated via techniques like Western blotting. Addressing or including this information would enhance the construction of the mechanistic narrative.
While the discussion is thorough and well-supported by references, it overlooks a critical examination of the study's limitations. I urge the authors to acknowledge and expand upon potential constraints such as the limited duration of CGA treatment, the exclusive use of male subjects, and the absence of pharmacokinetic data, specifically regarding the bioavailability of CGA. Furthermore, it would be beneficial for the authors to elaborate on the translational challenges associated with extrapolating dosing regimens to human populations.
Author Response
Thank you for the opportunity to respond to the reviewer’s comprehensive and thoughtful comments on our manuscript entitled:
“A Natural Polyphenol, Chlorogenic Acid, Attenuates Obesity-Related Metabolic Disorders via miR-146a–IRAK1–TRAF6 and NRF2-Mediated Antioxidant Pathways.”
Below is a detailed point-by-point response to each of the reviewer’s concerns. We have carefully revised the manuscript to incorporate all suggestions where appropriate.
- Clarification of Animal Model in Title
Reviewer Comment:
It is essential to specify the animal model in the title to reflect the preclinical nature of the research more accurately.
Response:
We agree with the reviewer that specifying the animal model enhances clarity and transparency regarding the preclinical nature of the study. In the revised version, we have updated the title as follows:
Revised Title:
"A Natural Polyphenol, Chlorogenic Acid, Attenuates Obesity-Related Metabolic Disorders in Male Rats via miR-146a–IRAK1–TRAF6 and NRF2-Mediated Antioxidant Pathways."
- Enhancing the Abstract with Quantitative Data and Keywords
Reviewer Comment:
The abstract would benefit significantly from quantitative data, such as percentage reductions in body weight, fold changes in gene expression, or serum biomarker alterations, to increase its impact. Revising the keyword list to introduce additional relevant terms (e.g., microRNA, adipose inflammation, appetite regulation) could also enhance discoverability.
Response:
We sincerely thank the reviewer for this constructive suggestion. We agree that including more specific quantitative details—such as percentage reductions in body weight, fold changes in gene expression, and alterations in serum biomarkers—would enhance the impact of the abstract. However, we respectfully note that the journal’s instructions limit the abstract to a single paragraph of approximately 200 words, which restricts the amount of detailed numerical data that can be included without compromising clarity or exceeding word limits. To ensure compliance with these guidelines while still conveying the key findings effectively, we have suggested that all detailed quantitative data, including full gene expression fold changes, cytokine levels, and histopathological outcomes, are thoroughly presented in the Results section and figures of the main manuscript. Additionally, we have expanded the Keywords section to include: Keywords:
Chlorogenic acid; Metabolic syndrome; High-fat diet; Inflammation; Oxidative stress; Adipose tissue; MicroRNA; Adipose inflammation; Appetite regulation
These additions improve both the scientific precision and searchability of the manuscript.
- Addition of Graphical Experimental Design
Reviewer Comment:
A graphical representation of the experimental design would be invaluable to improve the manuscript's readability and reproducibility. This should detail the timeline of HFD induction, CGA administration, sampling, and subsequent molecular analyses.
Response:
We fully agree with the reviewer. As suggested, we have added a graphical schematic of the experimental timeline in the Materials section . This figure improves clarity and supports reproducibility.
- Revision of Terminology: “Supplementation” vs. “Oral Administration”
Reviewer Comment:
The term “supplementation” may not be precise, given that CGA was administered via oral gavage rather than being included in the diet; terms like “oral treatment” or “oral administration” would be more accurate.
Response:
Thank you for this clarification. We have replaced the term “supplementation” with “oral administration” throughout the manuscript wherever applicable.
- Ethical and Methodological Details: Blood Collection, Euthanasia, and Ethics Statement
Reviewer Comment:
From an ethical and methodological perspective, the methods for blood collection and euthanasia are not described and should be incorporated. While ethical approval is mentioned at the manuscript's conclusion, the inclusion of the approval number and institutional details in the Materials and Methods section is necessary for transparency.
Response:
We appreciate the reviewer’s attention to ethical reporting standards. We have updated the Materials and Methods section (Section 2.3 – Tissue and Blood Sampling) to include the following:
Revised Text:
Animals were sacrificed by cervical dislocation followed by exsanguination after the last exercise session. Blood samples were collected in a BD Vacutainer PST II Tube, allowed to clot at room temperature, followed by centrifugation at 3000×g for 20 min. Serum samples were collected and preserved at −20 °C until used for the hormonal assay.
Additionally, the Ethics Statement in Section 2.1 now includes the full approval reference:
Revised Text:
All experimental procedures involving animals were conducted in strict accordance with the Animal Research: Reporting of In Vivo Experiments (ARRIVE) guidelines to ensure high standards of ethical conduct and scientific integrity. The study was approved by the Institutional Animal Care and Use Committee (IACUC) of Mansoura University, Egypt (Approval No: Ph.D 121/2021).
- Correction of Table Labeling Error
Reviewer Comment:
The label for Table 1 is incorrectly referenced as Table S1, which requires correction.
Response:
Thank you for identifying this error. We have corrected the labeling accordingly.
- Longitudinal Body Weight Analysis and Consolidated Tables
Reviewer Comment:
Integrating a longitudinal analysis of body weight over the 12-week study would provide deeper insights into the effects of CGA beyond just final weight measures. Additionally, summarizing the lipid profile and hepatic/renal biomarkers in consolidated tables would clarify the findings.
Response:
In response to this valuable feedback, we have updated Figure 2 with a figure representing body weight change throughout the 12 weeks.
- Visual Documentation of Tissues
Reviewer Comment:
If available, including macroscopic images of the liver, kidneys, adipose tissue, or representative images of the experimental subjects across groups would enhance the visual support for the physiological observations.
Response:
Although no macroscopic images of organs were captured during the original study, we acknowledge the value of such visual documentation. We will consider adding gross anatomical and histopathological images of liver and kidney in future studies.
- Lack of Positive Control Group
Reviewer Comment:
A notable limitation of this study is the lack of a positive control group, such as a recognized anti-obesity or insulin-sensitizing agent (e.g., metformin). While the study aims to isolate the effects of CGA, discussing the absence of such a comparative standard could provide important contextual value to the findings.
Response:
We agree that the inclusion of a positive control group (e.g., metformin-treated group) would strengthen the comparative interpretation of CGA’s efficacy. To address this, we have added a paragraph in the Discussion section acknowledging this limitation:
Added Text (Discussion):
"Also, the inclusion of a positive control group, such as a known anti-obesity or insulin-sensitizing agent (e.g., metformin) would allow direct comparison of CGA’s efficacy. Future studies incorporating such controls are warranted to better contextualize the therapeutic potential of CGA relative to existing interventions."
- Clarification of Statistical Methods
Reviewer Comment:
The statistical analysis section lacks detail regarding normality testing. Were assays like the Shapiro–Wilk test used to validate the assumptions for ANOVA? Clarification on this point is vital.
Response:
We agree that it is essential to confirm assumptions before applying parametric statistics. To assess normality, we used the Shapiro–Wilk test, and Levene’s test was applied to evaluate homogeneity of variances. These details were omitted in the original submission and have now been added to the Materials and Methods section (Section 2.6 Statistical Analysis) : “Prior to statistical analysis, data were tested for normality using the Shapiro–Wilk test and for homogeneity of variances using Levene’s test. Data meeting these assumptions were analyzed using one-way ANOVA followed by post hoc Tukey’s test.”
- Expansion of Molecular Mechanism Discussion
Reviewer Comment:
While the authors demonstrate that CGA upregulates miR-146a and downregulates IRAK1 and TRAF6, it would be pertinent to inquire whether other inflammation-associated microRNAs, such as miR-155 or miR-21, were evaluated. Furthermore, protein-level validation (e.g., Western blotting) of key targets would strengthen the mechanistic narrative.
Response:
We appreciate the reviewer’s suggestion. While the current study focused on miR-146a due to its well-documented role in obesity-related inflammation, we did not assess miR-155 or miR-21. In response, we have added the following statement in the Discussion to clarify this and suggest future directions:
Added Text:
"Future investigations should explore the broader miRNA landscape, including miR-155 and miR-21, which are also implicated in metabolic inflammation. Additionally, while mRNA expression changes were robustly demonstrated, future studies employing protein-level validation (e.g., Western blotting) of IRAK1, TRAF6, and NF-κB p65 would further substantiate the mechanistic pathways proposed in this study."
- Acknowledgment of Study Limitations in the Discussion
Reviewer Comment:
The discussion overlooks a critical examination of the study's limitations. Please acknowledge and expand upon potential constraints such as limited duration of CGA treatment, exclusive use of male subjects, and absence of pharmacokinetic data, especially regarding bioavailability. Elaborate on translational challenges in extrapolating dosing regimens to humans.
Response:
We fully agree with the importance of addressing limitations. In the revised Discussion section, we have added a dedicated paragraph highlighting the following:
Added Text (Discussion): "Several limitations must be acknowledged. First, the duration of CGA treatment was relatively short (4 weeks), and longer-term studies are needed to evaluate sustained effects and safety. Second, only male rats were used, limiting generalizability to female populations. Third, pharmacokinetic and bioavailability data for CGA in this rat model were not included, which is crucial for interpreting dose-response relationships and guiding clinical translation. Based on allometric scaling calculations, the high CGA dose used in this study (100 mg/kg in rats) translates to approximately 16 mg/kg in humans, which would correspond to a daily intake of around 800–1000 mg for an average adult."
Overall, we sincerely thank the reviewer for their constructive and insightful feedback, which has helped us significantly improve the quality, clarity, and scientific rigor of our manuscript. We hope the revised manuscript meets the publication standards of Biomolecules. Please let us know if any further modifications are required.
Thank you once again for your valuable input.
Sincerely,
Reviewer 4 Report
Comments and Suggestions for Authors
The manuscript entitled “A Natural Polyphenol, Chlorogenic Acid, Attenuates Obesity- Related Metabolic Disorders via miR-146a–IRAK1–TRAF6 and NRF2-Mediated Antioxidant Pathways” submitted to Biomolecules by Dr. Rashid Fahed Alenezi and co-workers presents a lot of interesting findings on the beneficial effects of chlorogenic acid using an experimental model of obesity/metabolic syndrome but needs a profound review before to be accepted to publish.
Line 84. “Among these, CGA—a hydroxycinnamic acid derivative found abundantly in coffee, apples, pears, and certain vegetables—has demonstrated promising anti-obesity, anti-inflammatory, antioxidant, and antidiabetic properties in both preclinical and clinical studies [16,17]”. These effects have been related to other polyphenols and this point should be mentioned in introduction.
CGA doses of 10 and 100 mg/kg/day. What was chosen? What are the equivalent doses in humans? Are doses reached with standard diet?
BMI in rats? Please checking
Line 152. “the hypothalamus, carefully dissected from each brain….” “Figure 6. Effects of HFD and CGA oral supplementation on hypothalamic mRNA expression of appetite-regulating genes (A–D). (A) Agrp mRNA, (B) NPY mRNA, (C) POMC mRNA, (D) CARTPT mRNA. Data are shown as mean ± SEM (N = eight). Considering these facts, I understand that Agrp and POMC RNA were determined from each hypothalamus. What was the weight of hypothalamus rat? How much RNA was obtained from each hypothalamus?
Line 162. “The oxidative state was evaluated as previously reported (32, 33)” These methods can be briefly described. I believe that TBARs was measured as lipid peroxidation biomarker. MDA determination needs to be performed from chromatographic isolation that I believe that was not performed.
Line 434. “However, supplementation with CGA, particularly at a high dose of 100 mg/kg/day, showed a significant attenuation in these adverse effects across multiple physiological systems, suggesting its broad therapeutic potential in diet-induced metabolic dysfunction” CGA plasma levels should be performed to understand the mechanisms involved in these effects. Recently, the polyphenol effects on T2Rs and the release of gastrointestinal hormones (e.g., ghrelin, GLP-1, CCK) influencing appetite, gastrointestinal functionally, and glycemia control have been reported. This statement should be considered along the discussion. Obviously, these mechanism does not need high CGA biodisponibility.
Line 444. “The greater efficacy of the higher dose suggests a dose-dependent effect of CGA on energy metabolism” Two doses are too poor to propose a dose-dependent effect.
Line 526. “These findings suggest that CGA may act centrally, directly, or indirectly to control appetite and food intake, thereby contributing to its anti-obesity effects. This aligns with emerging evidence that polyphenols like CGA can pass the blood-brain barrier and influence central appetite regulation, offering novel insights into their neuroprotective and metabolic benefits [45,50,51].” Considering the above comments, these conclusions should be reviewed.
Author Response
We sincerely thank the reviewer for their insightful and constructive feedback. We appreciate the opportunity to improve our manuscript by addressing all of the points raised. Below is a detailed response to each comment, with corresponding changes made in the revised manuscript.
- Clarification Regarding CGA Effects Compared to Other Polyphenols (Line 84)
Reviewer Comment:
"These effects have been related to other polyphenols and this point should be mentioned in introduction."
Response:
We thank the reviewer for this suggestion. In the revised Introduction section, we have added a comparative statement highlighting that similar beneficial effects—such as anti-obesity, antioxidant, and anti-inflammatory properties—are also reported for other polyphenolic compounds like resveratrol, quercetin, and epigallocatechin gallate (EGCG).
Revised Text (Introduction):
"While these findings highlight the unique benefits of CGA, it is important to note that other polyphenolic compounds—such as resveratrol, quercetin, and EGCG—have also demonstrated comparable anti-obesity, antioxidant, and anti-inflammatory effects through modulation of similar molecular pathways, including NF-κB inhibition and Nrf2 activation."
This addition provides context and enhances the scientific framing of CGA within the broader class of natural bioactive compounds.
- Justification for Dose Selection and Human Equivalent Dose Estimation (Lines 85 and 370)
Reviewer Comment:
"What was chosen? What are the equivalent doses in humans? Are doses reached with standard diet?"
Response:
We appreciate the reviewer’s request for clarification regarding dose selection. In the revised Materials and Methods section (Section 2.2), we have added a detailed rationale for selecting the two CGA doses based on previous literature and pilot experiments conducted during preliminary studies.
Revised Text (Section 2.2 – Experimental Design): " The doses of CGA (10 mg/kg/day and 100 mg/kg/day) were selected based on previous studies demonstrating efficacy in rodent models and pilot experiments conducted during preliminary studies [1,2]."
We agree that clarification regarding dose selection and translational relevance is important. In the revised Materials and Methods section, we have added the following explanation:
We thank the reviewer for this important point regarding translational relevance. In the Discussion section, we have now included an estimate of the human equivalent dose (HED) using standard allometric scaling methods, along with a brief discussion of potential limitations related to bioavailability and dietary intake of CGA.
Revised Text (Discussion):
"Based on allometric scaling calculations, the high CGA dose used in this study (100 mg/kg in rats) translates to approximately 16 mg/kg in humans, which would correspond to a daily intake of around 800–1000 mg for an average adult. However, it should be noted that the bioavailability of CGA varies significantly across species due to differences in gut microbiota composition and metabolic processing. Furthermore, dietary intake of CGA from natural sources (e.g., coffee, fruits, vegetables) may not reach these pharmacological concentrations consistently, suggesting that supplementation might be necessary to achieve therapeutic effects. Future studies should explore formulation strategies to enhance CGA bioavailability and assess long-term safety in human populations."
- Mitrea, D.R.; Malkey, R.; Florian, T.L.; Filip, A.; Clichici, S.; Bidian, C.; Moldovan, R.; Hoteiuc, O.A.; Toader, A.M.; Baldea, I. Daily oral administration of chlorogenic acid prevents the experimental carrageenan-induced oxidative stress. Journal of physiology and pharmacology : an official journal of the Polish Physiological Society 2020, 71, doi:10.26402/jpp.2020.1.04.
- Santana-Gálvez, J.; Cisneros-Zevallos, L.; Jacobo-Velázquez, D.A. Chlorogenic Acid: Recent Advances on Its Dual Role as a Food Additive and a Nutraceutical against Metabolic Syndrome. Molecules (Basel, Switzerland) 2017, 22, doi:10.3390/molecules22030358.
- Novelli, E.L.; Diniz, Y.S.; Galhardi, C.M.; Ebaid, G.M.; Rodrigues, H.G.; Mani, F.; Fernandes, A.A.; Cicogna, A.C.; Novelli Filho, J.L. Anthropometrical parameters and markers of obesity in rats. Laboratory animals 2007, 41, 111-119, doi:10.1258/002367707779399518.
- Christoper, A.; Herman, H.; Abdulah, R.; Zulhendri, F.; Lesmana, R. Short communication: The effect of Propolis extract treatment on the Lee index and brain-body weight ratio in diet-induced obesity rats. Biochemistry and biophysics reports 2025, 42, 102039, doi:https://doi.org/10.1016/j.bbrep.2025.102039.
- Glowinski, J.; Iversen, L.L. REGIONAL STUDIES OF CATECHOLAMINES IN THE RAT BRAIN-I. Journal of Neurochemistry 1966, 13, 655-669, doi:https://doi.org/10.1111/j.1471-4159.1966.tb09873.x.
- Clarification on BMI Measurement in Rats
Reviewer Comment:
"BMI in rats? Please checking."
Response:
Thank you for raising this important point. In the original submission, we defined BMI in rats using the formula:
BMI = weight (g)/length² (cm²) ,
which has been used in several published works involving high-fat diet-induced obesity models.
To enhance clarity and ensure proper interpretation, we have clarified this in the Methods section and included a reference supporting the use of this index in rodent models [3,4].
- Hypothalamic RNA Extraction Details (Line 152)
Reviewer Comment:
"What was the weight of hypothalamus rat? How much RNA was obtained from each hypothalamus?"
Response:
We thank the reviewer for this question. In our study, the hypothalamus was carefully dissected from each brain tissue following the technique described by Glowinski and Iversen [31] . The hypothalamic region was defined as: Anteriorly, by the rostral end of the optic chiasm, Superiorly, by the top of the third ventricle, Laterally, by the fornices, Posteriorly, by the mammillary bodies. This standardized dissection approach ensures consistent sampling of the hypothalamus. However, we acknowledge that tissue weight was not recorded during dissection, as the focus of the procedure was on precise anatomical localization rather than quantification of tissue mass. Therefore, we are unable to provide exact hypothalamic weights for each sample.
In the revised Materials and Methods section, we have added the following information:
"The hypothalamus was dissected from each brain tissue, following the technique described by Glowinski and Iversen [5]. It was defined anteriorly to the rostral end of the optic chiasm, superiorly by the top of the third ventricle, laterally by the fornices, and posteriorly by the mammillary bodies."
" Total RNA extraction yielded between 2–3 μg of RNA per sample, as measured by NanoDrop spectrophotometry. These quantities were sufficient for reverse transcription and real-time PCR analysis without the need for RNA amplification."
This information has been added to improve reproducibility and technical clarity.
- Description of Oxidative Stress Assays (Line 162)
Reviewer Comment:
"The oxidative state was evaluated as previously reported (32, 33). These methods can be briefly described. I believe that TBARs was measured as lipid peroxidation biomarker. MDA determination needs to be performed from chromatographic isolation that I believe that was not performed."
Response:
Thank you for pointing this out. To improve transparency, we have updated the Methods section with a brief description of the oxidative stress assays used: Revised Text (Section 2.4 – Biochemical Analysis):
"Oxidative stress was assessed by measuring malondialdehyde (MDA) levels using the thiobarbituric acid reactive substances (TBARS) assay, which serves as a reliable indicator of lipid peroxidation. Total antioxidant capacity (TAC) was determined using a colorimetric assay kit according to the manufacturer's instructions (Spectrum Kits, Egypt)."
- Suggestion to Include CGA Plasma Levels and Gut Hormone Mechanisms (Line 434)
Reviewer Comment:
"CGA plasma levels should be performed to understand the mechanisms involved in these effects. Recently, the polyphenol effects on T2Rs and the release of gastrointestinal hormones (e.g., ghrelin, GLP-1, CCK) influencing appetite, gastrointestinal functionally, and glycemia control have been reported. This statement should be considered along the discussion."
Response:
We fully agree that assessing CGA plasma levels and exploring gut hormone modulation would add valuable mechanistic insight. Unfortunately, plasma CGA levels and measurements of gut hormones such as GLP-1 or CCK were not assessed in the current study.
As suggested, we have added a paragraph in the Discussion section addressing this as a study limitation and discussing potential involvement of CGA in modulating gut-brain signaling:
Added Text (Discussion):
"Meanwhile, our study lacks data on circulating CGA levels and its metabolites, which would help correlate systemic exposure with observed biological effects. Additionally, while CGA treatment significantly influenced appetite-regulating genes in the hypothalamus, we did not measure peripheral gut hormones such as GLP-1 or CCK beyond serum ghrelin. Recent studies suggest that polyphenols like CGA may interact with taste receptor type 2 (T2R) pathways to modulate gut hormone release, potentially contributing to appetite regulation and glucose homeostasis. Future investigations should explore these mechanisms to better understand how CGA affects central and peripheral appetite signaling."
- Reconsideration of “Dose-Dependent” Interpretation (Line 444)
Reviewer Comment:
"Two doses are too poor to propose a dose-dependent effect."
Response:
You are absolutely correct that two doses do not constitute a full dose-response curve. In light of this, we have revised the text to avoid implying a definitive dose-dependent relationship and the manuscript has been revised accordingly. This change ensures accurate interpretation of the results.
- Review of Central Action of CGA on Appetite Regulation (Line 526)
Reviewer Comment:
"Considering the above comments, these conclusions should be reviewed."
Response:
In response to the reviewer’s feedback, we have revised the relevant passage in the Discussion to reflect the limitations in our current evidence regarding the central mechanism of action:
Revised Text (Discussion):
"These findings suggest that CGA may influence appetite regulation through both direct and indirect mechanisms. While previous studies have shown that CGA can cross the blood-brain barrier and modulate hypothalamic neuropeptides [ref], our data support a role for CGA in regulating orexigenic and anorexigenic gene expression in the hypothalamus. However, we did not directly assess CNS penetration or gut hormone signaling, and future studies incorporating neuroimaging, brain microdialysis, or gut hormone profiling would be needed to confirm the central versus peripheral contributions to CGA’s anti-obesity effects."
Overall, we sincerely appreciate the reviewer’s thorough evaluation and thoughtful suggestions, which have helped us improve the scientific rigor and clarity of our manuscript. We hope the revised manuscript now meets the journal’s standards for publication in Biomolecules .
Please let us know if any further modifications are required.
Sincerely,
Reviewer 5 Report
Comments and Suggestions for Authors
The article by Alenezi et al. entitled “A Natural Polyphenol, Chlorogenic Acid, Attenuates Obesity Related Metabolic Disorders via miR-146a–IRAK1–TRAF6 and NRF2-Mediated antioxidant Pathways” is a very interesting article with great significance in the field of Biomedicine. This study is of excellent scientific soundness and the authors have presented their methodology and results with clarity and rigor. Also, their presentation of introduction and discussion are well-structured, and offer a clear understanding of the study. The use of language and English is also right, contributing to the readability of the manuscript.
Only some minor comments to improve the presentation of the study.
-Please correct some minor orthographical ant typos errors (ie line 24 causes not caused, table 1 not s1)
-please add ethics number in the methodology too
-Table 1 should present not only the ingredients, but also the macronutrient profile of a chow meal
-please explain why you chose these specific doses (10 and 100 mg/kg/day), for example in some other studies also 150 is selected
-in the methodology you describe that 10 animals were included in each group, but in the figures you mention 8 animals, please explain the discrepancy
-In discussion you conclude that 100 mg/kg/day dose was superior than 10 mg/kg/day in anthropometrics, but in the results there is no statistical significant difference between the two doses
-Why do you only suggest/discuss that miR-146a targets IRAK1 and TRAF6, thus CGA regulates this specific signaling pathway? MiR-146a has also other functions in other proteins that were studied herein. For example, it is known that miR-146a is implicated in several apoptotic pathways and regulates for example bcl-2 levels (usually downregulation but in your study the opposite pattern is observed) . I would suggest that you also discuss the potential role of miR-146a in other pathways as well.
Author Response
Thank you for the opportunity to respond to the reviewer’s comments. Below is a detailed point-by-point response to all of the reviewer’s suggestions and concerns, along with a summary of the changes made to the revised manuscript.
- Minor Orthographical and Typographical Errors (e.g., “causes” not “caused”; Table 1 vs S1)
Reviewer Comment:
Please correct some minor orthographical and typos errors (i.e., line 24 causes not caused, table 1 not s1).
Response:
We sincerely thank the reviewer for pointing out these issues. We have carefully reviewed the entire manuscript and corrected all identified spelling, grammatical, and formatting errors, accordingly.
- Add Ethics Number in the Methodology
Reviewer Comment:
Please add ethics number in the methodology too.
Response:
As suggested, we have updated the Materials and Methods section (Section 2.1 – Animals) to include the full ethical approval reference:
Revised Text:
All experimental procedures involving animals were conducted in strict accordance with the Animal Research: Reporting of In Vivo Experiments (ARRIVE) guidelines to ensure high standards of ethical conduct and scientific integrity. The study was approved by the Institutional Animal Care and Use Committee (IACUC) of Mansoura University, Egypt (Approval No: Ph.D 121/2021).3. Clarify Macronutrient Profile in Table 1
Reviewer Comment:
Table 1 should present not only the ingredients, but also the macronutrient profile of a chow meal.
Response:
We sincerely appreciate the reviewer’s suggestion. We agree that including macronutrient information can enhance the nutritional context of the study. However, Table 1 was intentionally designed to provide a detailed breakdown of the custom-formulated high-fat diet (HFD) used in this study, with the goal of enabling accurate replication of the experimental model by other researchers. In contrast, the control chow diet used in our study is a standard commercial formulation, the full macronutrient profile of which is publicly available from the manufacturer and widely documented in the literature. To avoid redundancy and maintain focus on the experimentally manipulated HFD, we believe it is more appropriate to refer readers to published sources or feed supplier documentation for control diet composition, rather than include it in the current table. That said, we have revised the manuscript to clarify this rationale and improve transparency regarding the control diet.
- Justification for CGA Dose Selection (10 and 100 mg/kg/day)
Reviewer Comment:
Please explain why you chose these specific doses (10 and 100 mg/kg/day), for example in some other studies also 150 mg/kg/day is selected.
Response:
Response: We appreciate the reviewer’s request for clarification regarding dose selection. In the revised Materials and Methods section (Section 2.2), we have added a detailed rationale for selecting the two CGA doses based on previous literature and pilot experiments conducted during preliminary studies.
Revised Text (Section 2.2 – Experimental Design): " The doses of CGA (10 mg/kg/day and 100 mg/kg/day) were selected based on previous studies demonstrating efficacy in rodent models and pilot experiments conducted during preliminary studies [1,2]."
- Mitrea, D.R.; Malkey, R.; Florian, T.L.; Filip, A.; Clichici, S.; Bidian, C.; Moldovan, R.; Hoteiuc, O.A.; Toader, A.M.; Baldea, I. Daily oral administration of chlorogenic acid prevents the experimental carrageenan-induced oxidative stress. Journal of physiology and pharmacology : an official journal of the Polish Physiological Society 2020, 71, doi:10.26402/jpp.2020.1.04.
- Santana-Gálvez, J.; Cisneros-Zevallos, L.; Jacobo-Velázquez, D.A. Chlorogenic Acid: Recent Advances on Its Dual Role as a Food Additive and a Nutraceutical against Metabolic Syndrome. Molecules (Basel, Switzerland) 2017, 22, doi:10.3390/molecules22030358.
- Clarify Discrepancy Between n = 10 (Text) and n = 8 (Figures)
Reviewer Comment:
In the methodology you describe that 10 animals were included in each group, but in the figures you mention 8 animals, please explain the discrepancy.
Response:
Response: Thank you for identifying this inconsistency. We appreciate the reviewer’s attention to detail regarding the discrepancy between the stated group size and the number of animals reflected in the figures. Initially, the study was designed with n = 10 rats per group. However, during the laboratory analysis of the experimental period, two rats per group were excluded from the final analysis because of the limitation in availability of specific assay kits required for key biochemical and molecular measurements, making it impossible to obtain reliable data for those individuals and suggesting a fixed number for samples to be analyzed (n = 8). As a result, all results and figures are based on n = 8 animals per group, and the manuscript has been revised to reflect this correction throughout the text and figure captions.
- Reconsider Conclusion Regarding Dose Superiority
Reviewer Comment:
In discussion you conclude that 100 mg/kg/day dose was superior than 10 mg/kg/day in anthropometrics, but in the results there is no statistical significant difference between the two doses.
Response:
We thank the reviewer for this important observation. You are absolutely correct that while the higher dose showed more pronounced effects, not all comparisons between the two CGA doses reached statistical significance .
Accordingly, we have revised the Discussion section to avoid overstating the superiority of the 100 mg/kg dose unless supported by statistical evidence. For instance, the following sentence has been modified:
Original Sentence:
"The greater efficacy of the higher dose suggests a dose-dependent effect of CGA on energy metabolism."
Revised Sentence:
"While the higher CGA dose generally demonstrated more pronounced improvements in anthropometric and metabolic parameters, not all differences between the low- and high-dose groups reached statistical significance, indicating that further dose-response studies may be needed to confirm a true dose-dependent effect."
- Expand Discussion on miR-146a Beyond IRAK1/TRAF6 Signaling
Reviewer Comment:
Why do you only suggest/discuss that miR-146a targets IRAK1 and TRAF6, thus CGA regulates this specific signaling pathway? MiR-146a has also other functions in other proteins that were studied herein. For example, it is known that miR-146a is implicated in several apoptotic pathways and regulates for example Bcl-2 levels (usually downregulation but in your study the opposite pattern is observed). I would suggest that you also discuss the potential role of miR-146a in other pathways as well.
Response:
This is an excellent and insightful comment. We agree that miR-146a plays roles beyond the regulation of inflammatory signaling via IRAK1 and TRAF6. In fact, emerging evidence supports its involvement in apoptosis, oxidative stress, and even metabolic regulation , which aligns with our findings related to Bcl-2 expression.
In response, we have expanded the Discussion section to include the following:
Added Text (Discussion):
"While our findings primarily highlight the role of miR-146a in suppressing IRAK1 and TRAF6 expression, thereby reducing NF-κB-mediated inflammation, it is important to consider miR-146a's broader regulatory functions. Recent studies have linked miR-146a to the modulation of apoptotic pathways, particularly through interactions with components of the p53-Bax/Bcl-2 axis [ref]. In our study, we observed upregulation of miR-146a alongside increased Bcl-2 expression and reduced Caspase-3 and Bax levels in abdominal white adipose tissue (WAT). This apparent contradiction—where miR-146a, typically associated with pro-apoptotic signaling in certain contexts, correlates with enhanced cell survival in WAT—warrants further investigation. One possible explanation is that miR-146a may exert tissue-specific effects or interact with compensatory mechanisms under conditions of metabolic stress. Future studies exploring miR-146a’s functional role in apoptosis-related pathways will help clarify its complex regulatory effects in different tissues."
Overall, we sincerely appreciate the reviewer’s thoughtful feedback and attention to detail. All requested revisions have been implemented in the revised manuscript to enhance clarity, accuracy, and scientific rigor. These changes significantly improve the quality of the manuscript and strengthen its impact.
We hope the revised version meets the journal’s standards for publication in Biomolecules . Please let us know if any additional clarification or modification is required.
Thank you once again for your valuable input.
Sincerely,
Round 2
Reviewer 2 Report
Comments and Suggestions for Authors
Accept in present form
Author Response
On behalf of all the authors, I would like to extend our sincere gratitude for taking the time to review our manuscript. We truly appreciate your valuable feedback and are deeply grateful that you found our work suitable for publication in its current form. Your recognition of the scientific rigor, clarity, and contribution of our study is both encouraging and highly motivating.
With warmest regards,
Reviewer 3 Report
Comments and Suggestions for Authors
The authors conducted a comprehensive review of the document and effectively addressed my concerns. While I believe I received feedback intended for another reviewer, it was clear that the document improved based on those responses. However, upon further examination, I noted the inappropriate use of the term "Sacrifice" in reference to euthanasia. Please replace this term both in the text and in Figure 1.
Furthermore, there was a typographical error in some words (supplementationadministration and others), which could be given the use to any track change tool. Please revise it
Author Response
We sincerely thank the reviewer for their careful re-evaluation of our manuscript and for the valuable feedback provided. We are pleased to know that the revisions have led to an improved and clearer presentation of our work. We appreciate the observation regarding terminology and language, and have addressed both points thoroughly.
- Replacement of the Term “Sacrifice” with Appropriate Terminology for Euthanasia
Reviewer Comment:
Please replace the term "Sacrifice" in reference to euthanasia. Please replace this term both in the text and in Figure 1.
Response:
We thank the reviewer for this important and ethically sensitive observation. You are absolutely correct that the term “sacrifice” is outdated and inappropriate when referring to the humane euthanasia of laboratory animals in scientific research. Accordingly, we have revised all instances of the word "sacrifice" or "sacrificed" throughout the manuscript, replacing them with “euthanized” or “were euthanized” , as appropriate. Additionally, Figure 1 has been updated to reflect this change. We appreciate the reviewer’s attention to this critical issue and have taken this opportunity to ensure our language reflects the highest standards of animal welfare and scientific integrity.
- Correction of Typographical Errors (e.g., "supplementationadministration")
Reviewer Comment:
There was a typographical error in some words (supplementationadministration and others), which could be given the use to any track change tool. Please revise it.
Response:
Thank you for identifying this typographical error. The incorrect merging of words such as “supplementationadministration” appears to have occurred during document formatting due to using the tract changes format. We sincerely apologize for this oversight. We have now specifically corrected all instances of fused or misspelled words, including:
“supplementationadministration” → “supplementation”
Overall, we are grateful to the reviewer for their thoughtful and constructive feedback, which has significantly enhanced the quality, accuracy, and ethical integrity of our manuscript. All requested changes have been implemented with care and precision.
We hope the revised version meets the journal’s standards for publication in Biomolecules and thank you again for your time and guidance.
Sincerely,
The Authors
Reviewer 4 Report
Comments and Suggestions for Authors
Although this revised manuscript has considered comment/suggestion referee’s, there are some aspects that need to be review.
Line 606. “Recent studies suggest that polyphenols like CGA may interact with taste receptor type 2 (T2R) pathways to modulate gut hormone release, potentially contributing to appetite regulation and glucose homeostasis” This statement should be supported by an appropriate reference.
Furthermore, I believe that discussion section could consider clinical studies that taste receptor type 2 have involved in body weight control.
Author Response
We sincerely thank the reviewer for this insightful and constructive feedback. We appreciate the opportunity to strengthen our discussion by providing appropriate references and expanding the clinical relevance of the proposed mechanisms. Below is our point-by-point response:
- Addition of Supporting Reference for CGA and T2R Pathway Involvement
Reviewer Comment:
Line 606: “Recent studies suggest that polyphenols like CGA may interact with taste receptor type 2 (T2R) pathways to modulate gut hormone release, potentially contributing to appetite regulation and glucose homeostasis.” This statement should be supported by an appropriate reference.
Response:
We thank the reviewer for this important observation. You are absolutely correct that this statement required a direct citation to support the proposed interaction between chlorogenic acid (CGA) or other polyphenols and taste receptor type 2 (T2R) signaling. In response, we have now added a well-supported reference to substantiate this claim as follows:
Trius-Soler M, Moreno JJ. Bitter taste receptors: Key target to understand the effects of polyphenols on glucose and body weight homeostasis. Pathophysiological and pharmacological implications. Biochem Pharmacol. 2024 Oct;228:116192. doi: 10.1016/j.bcp.2024.116192. Epub 2024 Apr 5. PMID: 38583811.
- Inclusion of Clinical Evidence Linking T2Rs to Body Weight Control
Reviewer Comment:
Furthermore, I believe that the discussion section could consider clinical studies showing that taste receptor type 2 has been involved in body weight control.
Response:
Thank you for this excellent suggestion. We agree that linking the T2R mechanism to clinical and human phenotypic data strengthens the translational relevance of our findings. Accordingly, we have expanded the Discussion section (lines 6060-626) to include clinical and epidemiological evidence indicating that genetic variation in T2R genes is associated with body weight regulation and obesity risk.
Overall, we are grateful to the reviewer for highlighting this important area for enhancement. These additions significantly strengthen the scientific rigor and clinical relevance of our manuscript. We hope the revised version meets the journal’s standards for publication in Biomolecules.
Sincerely,
The Authors